# Transition Metal (Fe$_2$O$_3$, Co$_3$O$_4$ and NiO)-Promoted CuO-Based α-MnO$_2$ Nanowire Catalysts for Low-Temperature CO Oxidation

**Haiou Zhang** [1,†], **Yixin Zhang** [1,†], **Huikang Song** [1], **Yan Cui** [1], **Yingying Xue** [1], **Cai-e Wu** [2], **Chao Pan** [3,*], **Jingxin Xu** [3], **Jian Qiu** [4], **Leilei Xu** [1,*] and **Mindong Chen** [1,*]

1    Collaborative Innovation Centre of the Atmospheric Environment and Equipment Technology,
     School of Environmental Science and Engineering, Nanjing University of Information Science & Technology,
     Jiangsu Key Laboratory of Atmospheric Environment Monitoring and Pollution Control,
     Nanjing 210044, China
2    College of Light Industry and Food Engineering, Nanjing Forestry University, Nanjing 210037, China
3    State Environmental Protection Key Laboratory of Atmospheric Physical Modeling and Pollution Control,
     China Energy Science and Technology Research Institute Co., Ltd., Nanjing 210023, China
4    Jiangsu Shuang Liang Environmental Technology Co., Ltd., Jiangyin 214400, China
*    Correspondence: pcxyz@139.com (C.P.); leileixu88@gmail.com (L.X.); chenmdnuist@163.com (M.C.)
†    These authors contributed equally to this work.

**Abstract:** As a toxic pollutant, carbon monoxide (CO) usually causes harmful effects on human health. Therefore, the thermally catalytic oxidation of CO has received extensive attention in recent years. The CuO-based catalysts have been widely investigated due to their availability. In this study, a series of transition metal oxides (Fe$_2$O$_3$, Co$_3$O$_4$ and NiO) promoted CuO-based catalysts supported on the α-MnO$_2$ nanowire catalysts were prepared by the deposition precipitation method for catalytic CO oxidation reactions. The effects of the loaded transition metal type, the loading amount, and the calcination temperature on the catalytic performances were systematically investigated. Further catalyst characterization showed that the CuO/α-MnO$_2$ catalyst modified with 3 wt% Co$_3$O$_4$ and calcined at 400 °C performed the highest CO catalytic activity (T$_{90}$ = 75 °C) among the investigated catalysts. It was supposed that the loading of the Co$_3$O$_4$ dopant not only increased the content of oxygen vacancies in the catalyst but also increased the specific surface area and pore volume of the CuO/α-MnO$_2$ nanowire catalyst, which would further enhance the catalytic activity. The CuO/α-MnO$_2$ catalyst modified with 3 wt% NiO and calcined at 400 °C exhibited the highest surface adsorbed oxygen content and the best normalized reaction rate, but the specific surface area limited its activity. Therefore, the appropriate loading of the Co$_3$O$_4$ modifier could greatly enhance the activity of CuO/α-MnO$_2$. This research could provide a reference method for constructing efficient low-temperature CO oxidation catalysts.

**Keywords:** transition metal oxides; CuO-based catalysts; Co$_3$O$_4$ modification; α-MnO$_2$ nanowire; catalytic oxidation of CO



## 1. Introduction

As a toxic pollutant in the atmosphere, carbon monoxide (CO) has widely received public attention in recent years [1]. CO is not only mainly produced during the incomplete combustion of fossil fuels and motor vehicle emissions [2], but it is also a precursor of the ozone pollution [3], which can be harmful even at low concentrations. In recent years, scientists around the world have employed different methods for the removal of CO, such as photocatalysis [4], thermocatalysis oxidation [5], etc. Thermocatalysis oxidation is the most commonly used method for the oxidation of CO. The noble metal-based catalysts (such as Au [6,7], Pt [8–10], Pd [11], Ru [12,13], Rh [14] and Ag [15]), transition metal oxide based catalysts (such as MnO$_2$ [16], CeO$_2$ [17], ZrO$_2$ [18], Co$_3$O$_4$ [16]), and metal

hybrid catalysts (such as $MnCo_2O_4$ spinel [19], Ce-Zr solid solution [20]) can be used for the thermocatalytic oxidation of CO at low temperatures without generating the secondary contaminants [16]. Transition metal-based catalysts have good potential for application due to their low cost, high stability, and good activity [21,22]. Among the transition metal oxides, manganese dioxide ($MnO_2$) has many advantages, such as a low price, being environmentally friendly, and abundance in nature [2,23]. As is well known, $MnO_2$ usually exists in different crystalline phases with different structures, such as the α-, β-, and γ-types of the one-dimensional pore structure, the δ-type of the two-dimensional pore structure, and the λ-type of the three-dimensional network structure [24], which greatly depend on the different connectivity (corner-edge sharing) exhibited by the $[MnO_6]$ octahedra [25–28]. Liang et al. [28] prepared $MnO_2$ nanorods with four different crystalline phases for the oxidation of CO. It was found that the order of oxidation activity of different crystalline phases with the same nanorod morphology was greatly different, in decreasing order from α- ≈ δ- > γ- > β-$MnO_2$ (The temperature at which the conversion of CO on α-$MnO_2$ reached 100% was approximately 130 °C). This indicated that the oxidation activity of CO significantly depended on the phase structure and channel structure of the $MnO_2$. Tian et al. [29] investigated the effects of the crystalline phases of $MnO_2$ (α-, β-, and ε-$MnO_2$) on the performances of the oxidation of CO and toluene. It was found that the β-$MnO_2$ performed the highest activity for CO oxidation ($T_{90}$ = 75 °C) among the three crystalline phases of $MnO_2$, and the α-$MnO_2$ behaved with the lowest activity for CO oxidation ($T_{90}$ = 118 °C). The content of oxygen vacancy in the catalyst was also determined by in situ EPR spectra and the results showed that the catalytic activity of the catalyst was proportional to the concentration of oxygen vacancies, which were regarded as the active sites for the adsorption and dissociation of oxygen molecules. It was believed that the catalytic activity of $MnO_2$ was greatly related to the oxygen vacancy activity [29]. As is well known, the α-MnO2 is provided with the 1D (1 × 1) (2 × 2) tunnel structures, which are attributed to the tetragonal crystal system [30].

CuO has been widely used as the active site of the low-temperature catalytic oxidation of CO due to its excellent activity and abundant availability [31]. For example, Raziyeh Jokar et al. [31] used CuO/α-$MnO_2$ as the catalyst of the preferential oxidation of CO in the hydrogen-rich gas stream and investigated the interaction between the $MnO_2$ and CuO ($T_{97}$ = 130 °C), the superior activity of the catalyst due to the beneficial synergistic interaction between CuO and α-$MnO_2$. Meanwhile, the catalytic activity was also influenced by several factors, such as specific surface area, crystallinity, oxygen vacancies, and redox properties. Qian et al. [32] prepared a series of CuO/$MnO_2$ catalysts with different CuO loading amounts by the incipient wetness impregnation method for the oxidation of CO. The catalyst activity was almost the same for the CuO loadings, from 1% to 40%. Sun et al. [33] prepared a CuO/$Cu_{1.5}Mn_{1.5}O_4$ spinel-type composite oxide for synergistic catalysis of CO oxidation. It was found that the synergistic effect between $Cu_{1.5}Mn_{1.5}O_4$ spinel and CuO can promote the oxidation of CO, and CuO-$Cu_{1.5}Mn_{1.5}O_4$ had the best oxidation activity for CO ($T_{100}$ = 177 °C).

Nowadays, for the modification of transition metal oxide-based catalysts, in addition to the carrier and active site, the promoter also plays an active role in the improvement of catalytic performance [34]. Gao et al. [35] doped transition metals (Fe, Co, Ni, and Cu) with a 1:10 molar ratio on α-$MnO_2$ nanowires by a one-step hydrothermal method to oxidize CO. Among the four transition metals, $Cu_{0.1}MnO_x$ had the best oxidation activity for CO ($T_{100}$~120 °C). Krasimir et al. [36] investigated the effects of different molar ratios of chemical compositions on the γ-$Al_2O_3$-supported CuO/$MnO_2$/$Cr_2O_3$ catalysts for the oxidation of CO, dimethyl ether (DME), and methanol. The results showed that the Cu-Mn-Cr/γ-$Al_2O_3$ catalyst, which Cu/(Mn + Cr) has a molar ratio of 2:1 and a Mn/Cr molar ratio of 0.25, can achieve the complete oxidation of CO at 200 °C.

In order to further investigate the contribution of promoters to the catalytic performance of CuO/$MnO_2$ catalysts in the oxidation of CO, in this work, the α-$MnO_2$ nanowire was successfully prepared by the hydrothermal method and used as the support for the

CuO-based catalysts. A series of the transition metal oxides ($Fe_2O_3$, $Co_3O_4$, and NiO) promoted CuO-based $\alpha$-$MnO_2$ nanowire catalysts were prepared by the deposition precipitation method. The effects of the type, the loading amount, and the calcination temperature of three transition metal oxides on the performance of the catalytic oxidation of CO were systematically studied. The obtained catalysts were carefully characterized by X-ray diffraction (XRD), scanning electron microscopy (SEM), nitrogen physisorption, X-ray photoelectron spectroscopy (XPS), and $H_2$ temperature-programmed reduction ($H_2$-TPR). These catalysts were evaluated for their catalytic performances in the oxidation of CO. The results show that the $Co_3O_4$ (3 wt.%) promoted 10 wt.% CuO/$\alpha$-$MnO_2$ catalyst calcined at 400 °C performed the greatest CO reactivity ($T_{90}$ = 75 °C).

## 2. Results and Discussion

### 2.1. XRD Analysis

The crystalline phase structures of the catalysts were obtained using X-ray diffraction (XRD) analysis. The results of XRD are shown in the Figure 1. Figure 1a shows the XRD patterns of the transition metal oxide (MOx = $Fe_2O_3$, $Co_3O_4$, NiO) doped catalysts and the pristine 10CuO/$\alpha$-$MnO_2$-400 catalyst. It can be observed in Figure 1a that all the catalysts show wide and clear diffraction peaks at $2\theta$ = 12.78°, 18.11°, 25.71°, 37.52°, 41.97°, 49.86°, 56.37°, 65.11°, and 69.71°, which could be conformed to the characteristic peaks of the $\alpha$-$MnO_2$ (PDF#44-0141). It could be observed that the intensity of the diffraction peaks of $MnO_2$ increased after the loading of the second transition metal oxides, especially over the 10CuO-3$Co_3O_4$/$\alpha$-$MnO_2$ catalyst. This phenomenon suggests that the crystallinity of the catalysts increased after loading the transition metal oxides, especially for the catalyst loading $Co_3O_4$. Meanwhile, two weak diffraction peaks could be detected at $2\theta$ = 35.5° and 38.8°, which corresponded to the characteristic peaks of CuO, according to the standard card of PDF#05-0661. The diffraction peak intensity of CuO became weaker after the addition of the transition metal to the catalyst, indicating that the addition of the transition metal promoted the dispersion of CuO.

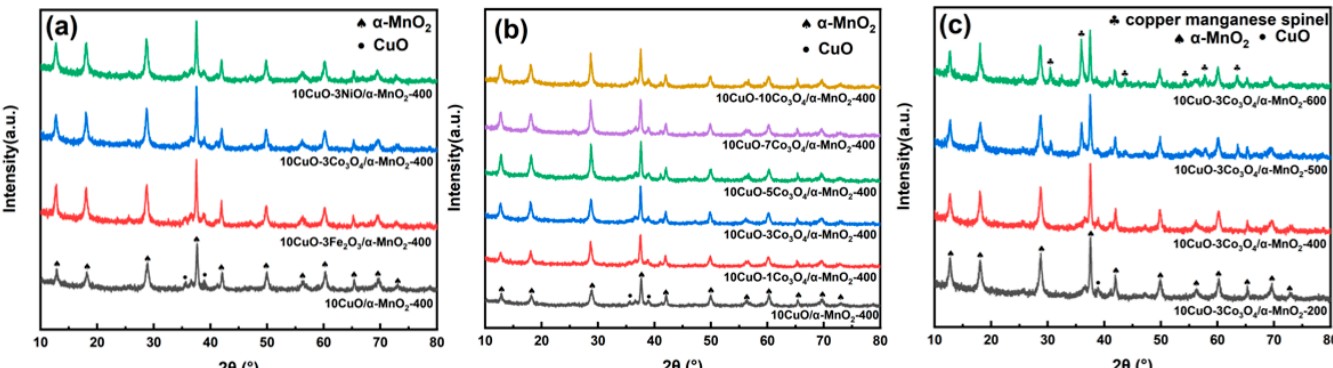

**Figure 1.** Powder XRD patterns of the (**a**) 10CuO-3MOx/$\alpha$-$MnO_2$-400 catalysts doped with different transition metals (MOx = $Fe_2O_3$, $Co_3O_4$, NiO); (**b**) 10CuO-y$Co_3O_4$/$\alpha$-$MnO_2$-400 catalysts doped with different contents of $Co_3O_4$ (y = 1, 3, 5, 7, 10); (**c**) 10CuO-3$Co_3O_4$/$\alpha$-$MnO_2$-T catalysts calcined at different temperatures (T = 200, 400, 500, 600 °C).

The XRD patterns of the 10CuO-y$Co_3O_4$/$\alpha$-$MnO_2$-400 catalysts with different contents of $Co_3O_4$ are shown in the Figure 1b. It can be observed that the diffraction peaks of CuO are very clear over the pristine 10CuO/$\alpha$-$MnO_2$-400 catalyst without $Co_3O_4$ loading. However, it can be observed that the diffraction peaks of CuO became blurry after loading $Co_3O_4$ from 1 wt.% to 10 wt.%. The possible reason accounting for this phenomenon was that the appropriate loading amount of $Co_3O_4$ could greatly enhance the dispersion of CuO on the surface of $\alpha$-$MnO_2$ nanowire catalyst. Nevertheless, the characteristic peaks of $Co_3O_4$ could be not observed in the catalysts due to the good dispersion of $Co_3O_4$.

The XRD patterns of $10CuO-3Co_3O_4/\alpha-MnO_2$ catalysis calcined at different temperatures are shown in the Figure 1c. It can be recognized that the intensity of the diffraction peaks of $\alpha-MnO_2$ became weak at high calcination temperatures. The reason might be due to the formation of a copper manganese spinel (JCPDS No.70-0260) [37]. Only a diffraction peak of CuO was observed at 38.8°. Meanwhile, the intensity of the diffraction peaks of CuO became sharp with the increase of the calcination temperature (only from 200 °C to 400 °C). This indicated that the dispersion of CuO on the catalyst surface gradually deteriorated and the crystal size of CuO nanoparticles grew due to the finite surface area of the $MnO_2$ nanowire from 200 °C to 400 °C. As the calcination temperature rose from 400 °C to 600 °C, the diffraction peak strength of copper manganese spinel increased. The diffraction peak strength of CuO decreased, indicating that the dispersion of CuO increased.

## 2.2. SEM Observation

The morphologies of the $10CuO-3MOx/\alpha-MnO_2-400$ catalysts loaded with different transition metal oxides (MOx = $Fe_2O_3$, $Co_3O_4$, NiO) and the pristine $10CuO/\alpha-MnO_2-400$ catalyst were characterized by SEM (Figure 2). It can be observed that the morphology of the $\alpha-MnO_2$ nanowire support greatly changed after loading the CuO active sites and the transition metal oxides. Specifically, the length to diameter ratio of the $\alpha-MnO_2$ nanowire significantly decreased compared to the pristine $\alpha-MnO_2$ nanowire. This was caused by the high loading contents of the CuO and transition metal oxides. In addition, the $\alpha-MnO_2$ nanowire support might experience thermal sintering and self-assemble at high calcination temperatures. The spatial distribution of the elements over the one-dimensional $10CuO-3MOx/\alpha-MnO_2-400$ (MOx = $Fe_2O_3$, $Co_3O_4$, NiO) nanowire catalyst was studied by energy dispersive X-ray spectroscopy mapping (EDS-mapping). As shown in Figure 2, the Cu and the doped Fe/Co/Ni elements were homogenously distributed over these investigated catalysts. This indicated that the great dispersion of the CuO and the doped transition metal oxides on the surface of the $\alpha-MnO_2$ nanowire supports could be facilely achieved by the precipitation deposition method.

## 2.3. BET Analysis

In order to further investigate the structural properties of the catalysts, the specific surface areas, pore volumes, and pore size distributions of the catalysts were measured by nitrogen physisorption measurements. As shown in Figure 3a, all catalysts show IV isotherms with H3-shaped hysteresis loops. These results prove the presence of mesopores with a narrow slit-shape in the catalyst [38]. It is also interesting to find that the $10CuO-3MOx/\alpha-MnO_2$ catalysts still possess the similar mesoporous structure to the $10CuO/\alpha-MnO_2$ nanowire catalysts after loading different transition metal oxides. This demonstrates that the $10CuO/\alpha-MnO_2$ nanowire catalyst was provided with good thermal stability. The pore size distribution curves of the corresponding catalysts are shown in Figure 3b. The pore diameter of the catalysts is in the range of 2–15 nm after loading with transition metals. Moreover, the specific parameters of the structural properties of these catalysts are shown in Table 1. It can be observed that the specific surface areas of both the $10CuO-3Fe_2O_3/\alpha-MnO_2$ and $10CuO-3NiO/\alpha-MnO_2$ catalysts decreased after the loading of transition metals. On the contrary, the specific surface area of the $10CuO-3Co_3O_4/\alpha-MnO_2$ catalysts increased substantially. This indicates that the $Co_3O_4$ on the $\alpha-MnO_2$ nanowire support surface was highly dispersed. Meanwhile, the average pore sizes of all the $10CuO-3MOx/\alpha-MnO_2$ (MOx = $Fe_2O_3$, $Co_3O_4$, NiO) catalysts are very similar to the pristine $10CuO/\alpha-MnO_2$ nanowire. Specifically, the $10CuO-3Fe_2O_3/\alpha-MnO_2$ catalyst has a similar pore size distribution as the $10CuO/\alpha-MnO_2$ catalyst. This indicates that their mesoporous structures are not sharply impaired by the loaded metal oxides. In addition, the pore capacities of all $10CuO-3MOx/\alpha-MnO_2$ (MOx = $Fe_2O_3$, $Co_3O_4$, NiO) catalysts are enhanced compared to the pristine $10CuO/\alpha-MnO_2$. As for the $10CuO-3Co_3O_4/\alpha-MnO_2$ catalyst, its surface area was twice as large as the pristine $10CuO/\alpha-MnO_2$ catalyst. The higher specific surface area is beneficial for the catalyst to expose more active sites, and the larger pore

volume helps the reactant accelerate the reactant mass diffusion and has a better adsorption ability to the reactant [39].

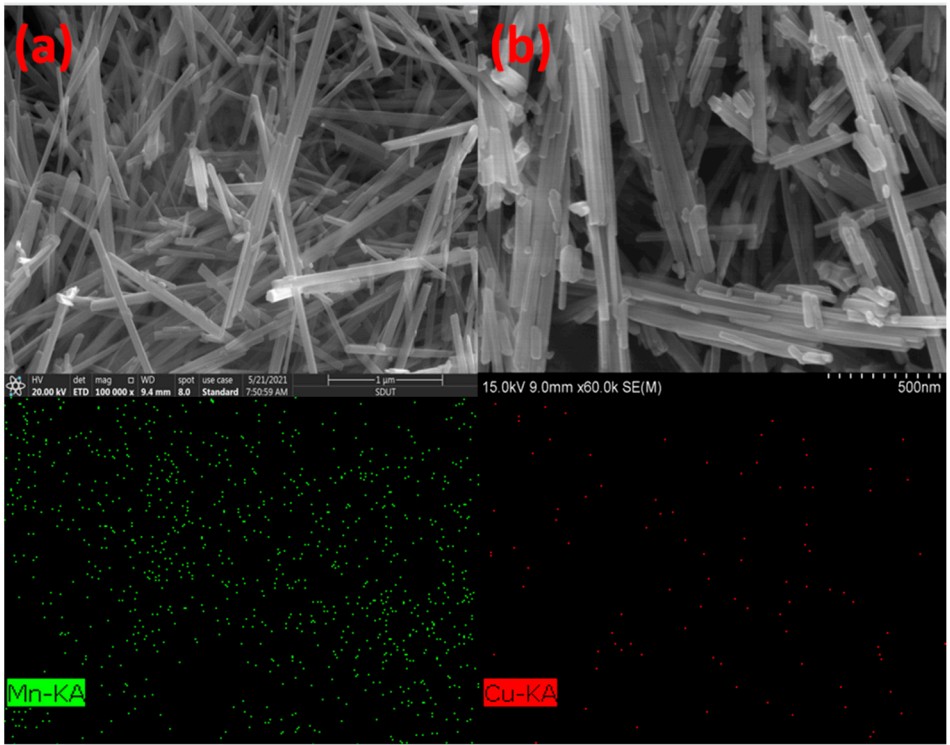

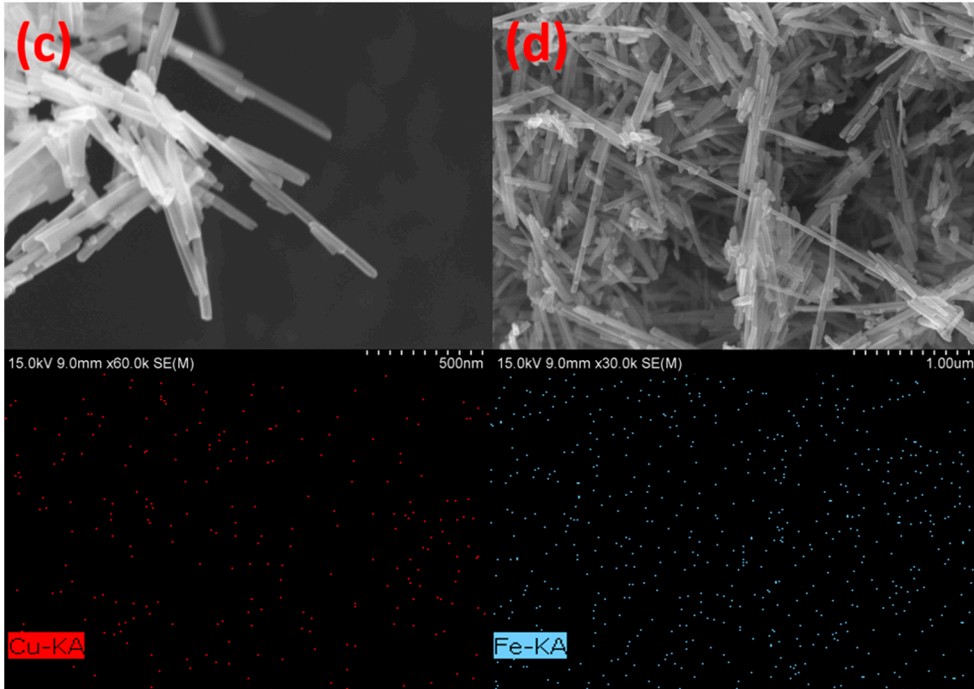

**Figure 2.** *Cont.*

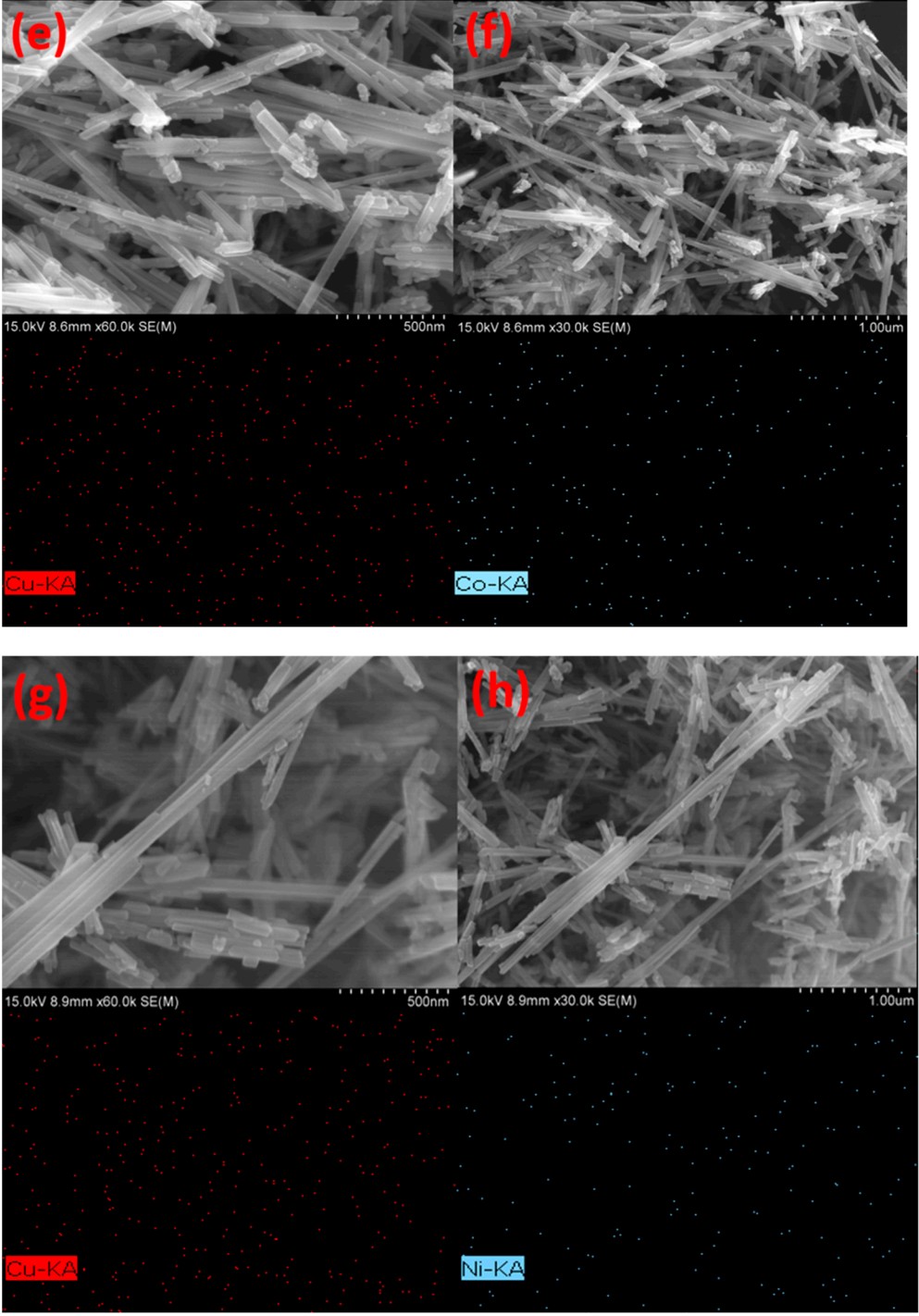

**Figure 2.** SEM-EDS images of the (**a**) $\alpha$-MnO$_2$ nanowire, (**b**) 10CuO/$\alpha$-MnO$_2$-400, (**c**,**d**) 10CuO-3Fe$_2$O$_3$/$\alpha$-MnO$_2$-400, (**e**,**f**) 10CuO-3Co$_3$O$_4$/$\alpha$-MnO$_2$-400, and (**g**,**h**) 10CuO-3NiO/$\alpha$-MnO$_2$-400.

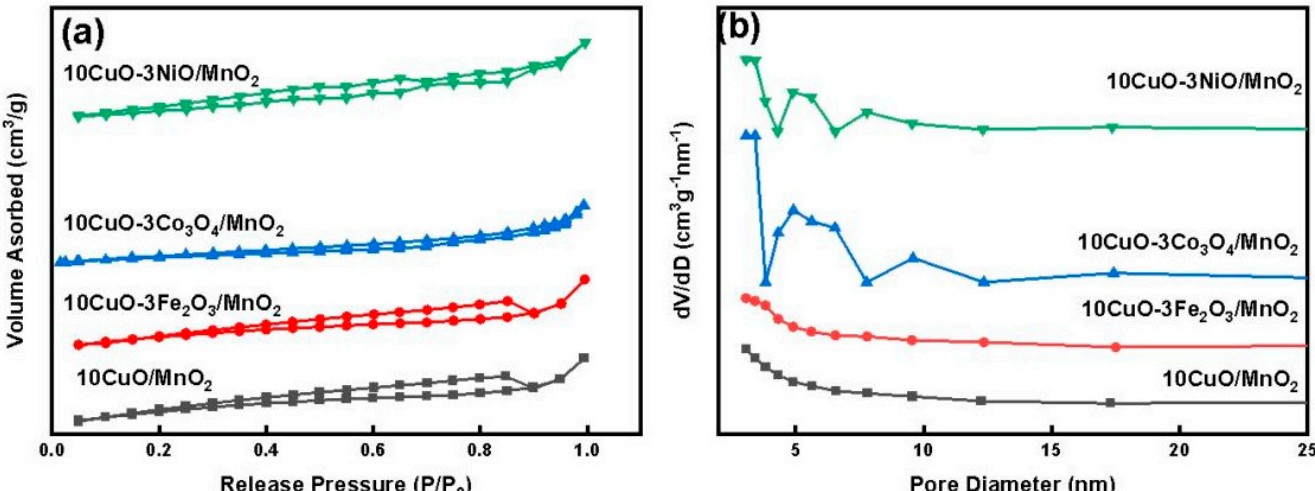

**Figure 3.** (**a**) Nitrogen adsorption-desorption isotherms and (**b**) BJH pore size distribution of the 10CuO/$\alpha$-MnO$_2$-400 and 10CuO-3MOx/$\alpha$-MnO$_2$-400 (MOx = Fe$_2$O$_3$, Co$_3$O$_4$, NiO) catalysts.

**Table 1.** Structural properties of the 10CuO/$\alpha$-MnO$_2$-400 and 10CuO-3MOx/$\alpha$-MnO$_2$-400 (MOx = Fe$_2$O$_3$, Co$_3$O$_4$, NiO) catalysts.

| Catalyst | Specific Surface Area (m$^2$/g) | Pore Volume (cm$^3$/g) | Average Pore Diameter (nm) | Isotherm Type |
|---|---|---|---|---|
| 10CuO/MnO$_2$ | 72 | 0.09 | 3.1 | IV H3 |
| 10CuO-3Fe$_2$O$_3$/MnO$_2$ | 64 | 0.10 | 3.1 | IV H3 |
| 10CuO-3Co$_3$O$_4$/MnO$_2$ | 84 | 0.18 | 3.1 | IV H3 |
| 10CuO-3NiO/MnO$_2$ | 52 | 0.10 | 3.1 | IV H3 |

*2.4. XPS Analysis*

The coordination, composition, and valence state of the elements over the catalyst surface were investigated by X-ray photoelectron spectroscopy (XPS). The XPS of Mn 2p, O 1s, and Cu 2p of the 10CuO/$\alpha$-MnO$_2$-400 and the 10CuO-3MOx/$\alpha$-MnO$_2$-400 (MOx = Fe$_2$O$_3$, Co$_3$O$_4$, NiO) are shown in Figure 4. It can be observed in Figure 4a that 10CuO/$\alpha$-MnO$_2$-400 had two main peaks around 654.0 eV and 642.0 eV, which could be attributed to the binding energies of the Mn 2p$_{1/2}$ at BE = 654.0 eV and Mn 2p$_{3/2}$ at BE = 642.0 eV, respectively. The two main peaks were divided into four peaks after the peak fitting. The fitted peaks of Mn 2p$_{3/2}$ at 642.0 eV and 643.5 eV indicate the existence of Mn$^{3+}$ and Mn$^{4+}$ in the 10CuO-3MOx/$\alpha$-MnO$_2$-400 catalyst [40–44]. The ratios of the Mn$^{3+}$/(Mn$^{3+}$ + Mn$^{4+}$) over different catalysts followed the below order: 10CuO-3Co$_3$O$_4$/$\alpha$-MnO$_2$ (0.606) > 10CuO-3NiO/$\alpha$-MnO$_2$ (0.541) > 10CuO/$\alpha$-MnO$_2$ (0.414) > 10CuO-3Fe$_2$O$_3$/$\alpha$-MnO$_2$ (0.406). The redox electron pair in Cu-Mn oxide was the -Cu$^{2+}$-O$^{2-}$-Mn$^{4+}$-$\rightarrow$-Cu$^+$-$\square$-Mn$^{3+}$- + 1/2O$_2$ ($\square$ indicates the oxygen vacancy) [45]. The content of Mn$^{3+}$ and oxygen vacancies are proportional, or indirectly proportional, to the oxidation capacity of the catalyst [25,29,46,47]. The Mn$^{3+}$ may cause the Jahn-Teller effect, which could prolong the Mn-O bond in [MnO$_6$] [48,49], thereby prolonging the distance between the oxygen pairs and causing the stretching of the Mn-O bond length [48]. As a result, the Mn-O bond was easier to break, and the mobility of oxygen became higher. Therefore, the released surface oxygen atoms are more likely to participate in the reaction and thus promote the catalytic performance. The XPS of O 1s of 10CuO-3MOx/$\alpha$-MnO$_2$-400 was measured to elucidate the nature of the oxygen species over the 10CuO-3MOx/$\alpha$-MnO$_2$-400 catalysts. As shown in Figure 4b, all the samples show double peaks of different oxygen species. Specifically, the BEs at around 529.8 eV and 531.4 eV could be ascribable to the surface lattice oxygen (O$_{latt}$) and surface adsorbed oxygen (O$_{ads}$) species [50,51], respectively. The ratios of the

$O_{ads}/O_{latt}$ are also summarized in Table 2. It can be observed that the binding energy of surface lattice oxygen ($O_{latt}$) shifted to a higher binding energy with the addition of transition metals. The surface oxidation states of the Cu species were also investigated to show the redox properties of the as-prepared catalysts. As shown in Figure 4c, all the catalysts displayed two main peaks of Cu $2p_{1/2}$ (953.6 eV) and Cu $2p_{3/2}$ (933.7 eV) [43]. The Cu $2p_{3/2}$, orbitals with binding energy in the range of 930.0–935.0 eV, could be divided into two peaks. Specifically, the binding energy peak at 933.4 eV was attributed to $Cu^+$, and the peak at 934.4 eV was attributed to $Cu^{2+}$ [43]. Furthermore, it is worth noting that the Cu $2p_{3/2}$ peak is accompanied by the vibrating satellite peaks in the range of 940.0–944.0 eV [52]. Combined with the results of the Mn 2p spectrum, the catalysts formed redox pairs of $Cu^+/Cu^{2+}$ and $Mn^{3+}/Mn^{4+}$, which would promote the charge transference to generate more oxygen defects [37,53].

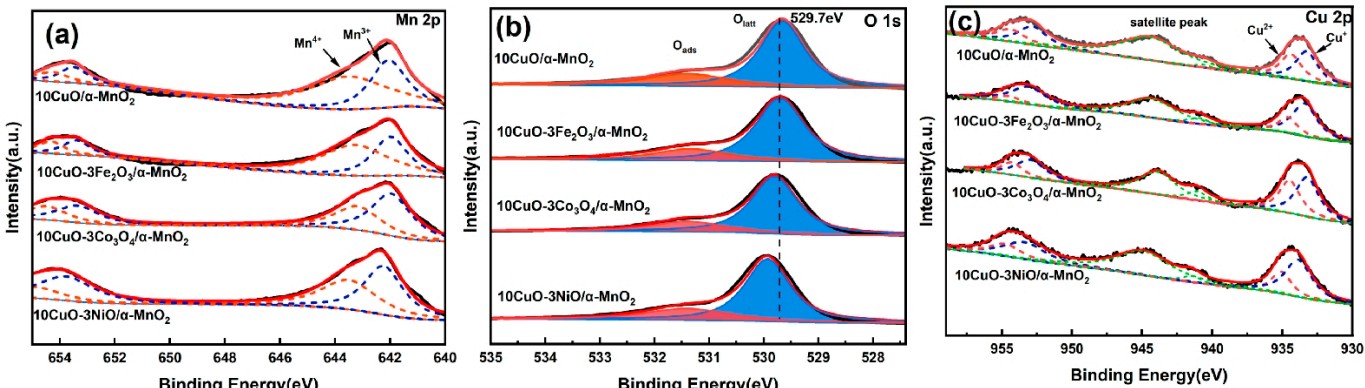

**Figure 4.** (**a**) Mn 2p spectra, (**b**) O 1s spectra, and (**c**) Cu 2p spectra of the 10CuO/$\alpha$-MnO$_2$, 10CuO-3Fe$_2$O$_3$/$\alpha$-MnO$_2$, 10CuO-3Co$_3$O$_4$/$\alpha$-MnO$_2$, and 10CuO-3NiO/$\alpha$-MnO$_2$ (T = 400 $^{\circ}$C) catalysts.

**Table 2.** The ratio of $Mn^{3+}/(Mn^{3+} + Mn^{4+})$, $O_{ads}/O_{latt}$, and $Cu^+/(Cu^{2+} + Cu^+)$ of the 10CuO-3MOx/$\alpha$-MnO$_2$(MOx = Fe$_2$O$_3$, Co$_3$O$_4$, NiO) and the 10CuO/$\alpha$-MnO$_2$ catalyst.

| Sample | $Mn^{3+}/(Mn^{3+} + Mn^{4+})$ | $O_{ads}/O_{latt}$ | $Cu^+/(Cu^{2+} + Cu^+)$ |
|---|---|---|---|
| 10CuO/$\alpha$-MnO$_2$ | 0.414 | 0.324 | 0.551 |
| 10CuO-3Fe$_2$O$_3$/$\alpha$-MnO$_2$ | 0.406 | 0.283 | 0.644 |
| 10CuO-3Co$_3$O$_4$/$\alpha$-MnO$_2$ | 0.606 | 0.300 | 0.563 |
| 10CuO-3NiO/$\alpha$-MnO$_2$ | 0.541 | 0.352 | 0.688 |

To determine the valence state of the transition metals loaded on the catalyst, the XPS spectra of Fe 2p, Co 2p, and Ni 2p are determined. The XPS profile of Fe 2p over the 10CuO-3Fe$_2$O$_3$/$\alpha$-MnO$_2$-400 catalyst is shown in Figure 5a. The binding energies at 710.4 eV and 725.1 eV are ascribed to Fe $2p_{3/2}$ and Fe $2p_{1/2}$, respectively [54,55]. The peak of Fe $2p_{3/2}$ can be divided into two peaks (710.3 eV and 712.5 eV) [56]. In addition, a satellite peak was observed at about 718.3 eV. This indicates that the iron species existed in the form of $Fe^{3+}$ on the surface of the 10CuO-3Fe$_2$O$_3$/$\alpha$-MnO$_2$-400 [54,56]. The XPS of Co 2p over the 10CuO-3Co$_3$O$_4$/$\alpha$-MnO$_2$-400 catalyst is shown in Figure 5b. The binding energies at 780.0 eV are ascribed to the Co $2p_{3/2}$ [57]. Meanwhile, there was no significant satellite shake-up intensity at 786 eV, indicating the dominance of $Co^{3+}$ on the surface of Co$_3$O$_4$ [57]. The XPS of Ni 2p over the 10CuO-3NiO/$\alpha$-MnO$_2$-400 catalyst is shown in Figure 5c. The binding energies at 855.1eV are ascribed to the Ni $2p_{3/2}$ [58]. The peak of Ni $2p_{3/2}$ of metallic Ni was basically at 852.6 eV, and the binding energy of NiO is about 1 eV higher than that of $Ni^0$ [58]. The peak of NiO $2p_{3/2}$ is lower than that of Ni $2p_{3/2}$ in this work (854.8 eV). Therefore, the oxidation state of the indicated Ni element is mainly in the form of $Ni^{2+}$. The higher binding energy compared to pure NiO binding energy indicates that NiO did not exist in the free form. The strong interaction was formed between the

Ni and the support. The results indicate that $Fe_2O_3$, $Co_3O_4$, and NiO were successfully loaded on the catalyst. The XPS electronic binding energies of the surface elements of the 10CuO-3MOx/$\alpha$-MnO$_2$-400 (MOx = $Fe_2O_3$, $Co_3O_4$, NiO) are summarized in Table 3.

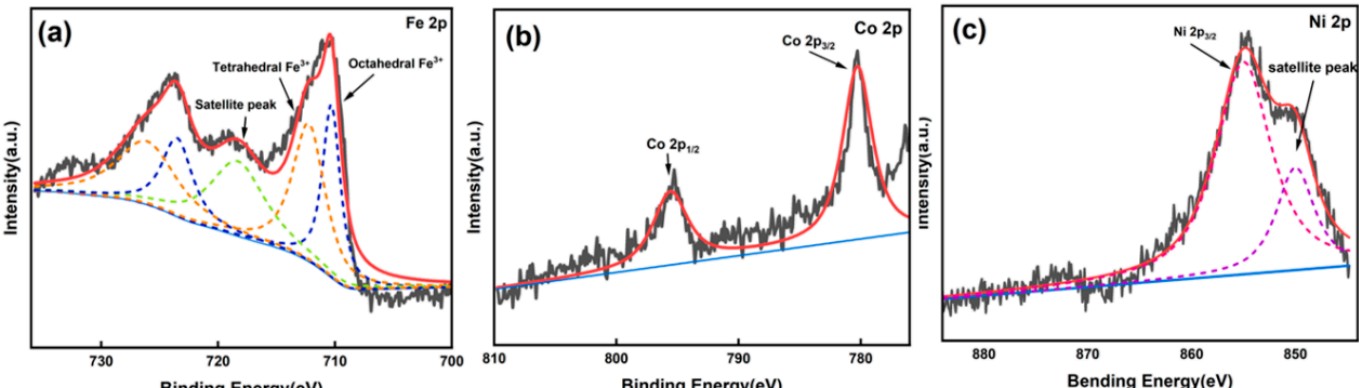

**Figure 5.** (**a**) Fe 2p, (**b**) Co 2p, and (**c**) Ni 2p spectra of the 10CuO-3MOx/$\alpha$-MnO$_2$-400 (MOx = $Fe_2O_3$, $Co_3O_4$, NiO) catalysts.

**Table 3.** XPS electronic binding energies of surface elements of the 10CuO-3MOx/$\alpha$-MnO$_2$-400 (MOx = $Fe_2O_3$, $Co_3O_4$, NiO) catalysts.

| Sample | Cu 2p$_{3/2}$ | O 1s | Mn 2p$_{3/2}$ |
|---|---|---|---|
| 10CuO/$\alpha$-MnO$_2$ | 933.6 | 529.7 | 642.1 |
| 10CuO-3Fe$_2$O$_3$/$\alpha$-MnO$_2$ | 933.7 | 529.7 | 642.1 |
| 10CuO-3Co$_3$O$_4$/$\alpha$-MnO$_2$ | 933.7 | 529.8 | 642.3 |
| 10CuO-3NiO/$\alpha$-MnO$_2$ | 934.2 | 529.9 | 642.4 |

### 2.5. H$_2$-TPR

The H$_2$-TPR profiles of 10CuO-3MOx/$\alpha$-MnO$_2$-400 (MOx = $Fe_2O_3$, $Co_3O_4$, NiO), 10CuO-yCo$_3$O$_4$/$\alpha$-MnO$_2$-400 (y = 0, 1, 3, 5, 7, 10), and 10CuO-3Co$_3$O$_4$/$\alpha$-MnO$_2$-T (T = 200, 400, 500, 600 °C) are shown in Figure 6. It can be observed that both the 10CuO-3MOx/$\alpha$-MnO$_2$-400 catalyst and the 10CuO/$\alpha$-MnO$_2$-400 catalyst show similar hydrogen consumption peaks in strength and shape in Figure 6a. Specifically, there were two sets of peaks in the range of 297–342 °C and 462–472 °C, which might be attributed to the hydrogen consumption derived from the gradual reduction of the $\alpha$-MnO$_2$ nanowire (MnO$_2$→Mn$_2$O$_3$→Mn$_3$O$_4$), according to the pioneer report [31]. Meanwhile, it is worth noting that the loading of the transition metal oxides on the 10CuO/$\alpha$-MnO$_2$ support changed the reduction behavior of the 10CuO/$\alpha$-MnO$_2$ catalyst. Specifically, the reduction of 10CuO/$\alpha$-MnO$_2$ nanowires shifted to higher temperatures after loading the transition metal oxides. In general, the reducibility of the 10CuO/$\alpha$-MnO$_2$ catalyst decreases with the addition of transition metal oxides. The H$_2$-TPR profiles of the 10CuO-yCo$_3$O$_4$/$\alpha$-MnO$_2$-400 catalysts with different Co$_3$O$_4$ loading amounts are shown in Figure 6b. It is of great interest to find that the positions of the reduction peaks gradually shifted to the high-temperature region with the Co$_3$O$_4$ loading amount increasing from 1 wt% to 10 wt%. This illustrated that the reducibility of the catalysts also gradually decreases. Therefore, the reduction temperatures of the 10CuO-yCo$_3$O$_4$/$\alpha$-MnO$_2$-400 catalyst were significantly higher than that of the 10CuO/$\alpha$-MnO$_2$ catalyst, except for the 10CuO-7Co$_3$O$_4$/$\alpha$-MnO$_2$-400 catalyst. The H$_2$-TPR profiles of the 10CuO-3MOx/$\alpha$-MnO$_2$-T catalyst with different calcination temperatures is shown in Figure 6c. It can be observed that the reduction temperature of the 10CuO-3Co$_3$O$_4$/$\alpha$-MnO$_2$-T catalyst gradually shifted to a higher reduction temperature with the increase of the calcination temperature from 200 °C to 600 °C. The positions of the two reduction peaks became closer. This suggests that the interaction between the CuO and the $\alpha$-MnO$_2$ nanowire support became stronger at higher temperatures.

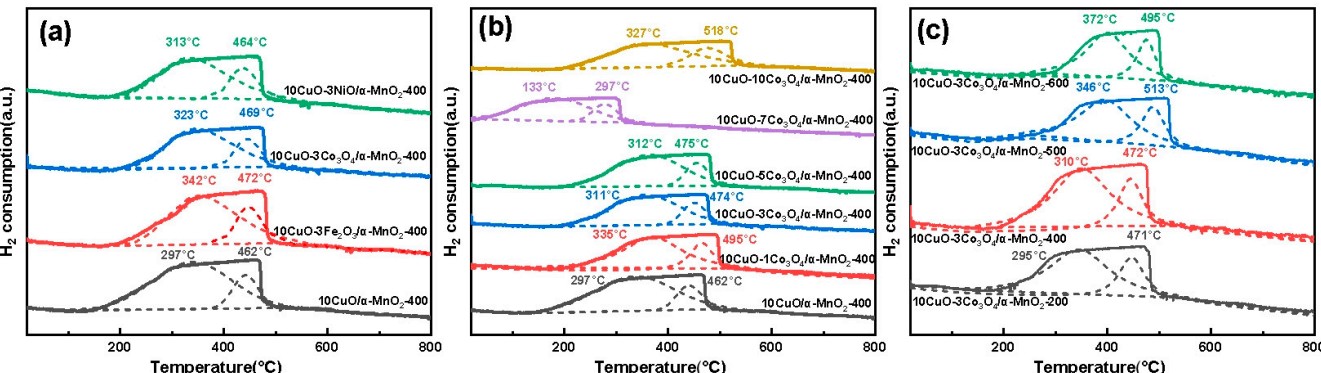

**Figure 6.** $H_2$-TPR profiles of the (**a**) 10CuO-3MOx/$\alpha$-MnO$_2$-400 (MOx = Fe$_2$O$_3$, Co$_3$O$_4$, NiO), (**b**) 10CuO-yCo$_3$O$_4$/$\alpha$-MnO$_2$-400 (y = 0, 1, 3, 5, 7, 10), and (**c**) 10CuO-3Co$_3$O$_4$/$\alpha$-MnO$_2$-T (T = 200, 400, 500, 600 °C) catalysts.

## 2.6. Catalytic Performance of the CO Oxidation

### 2.6.1. The Effect of the Transition Metal Oxides (MOx) on the Activities

The catalytic CO oxidation activities of the 10CuO-3MOx/$\alpha$-MnO$_2$-400 (MOx = Fe$_2$O$_3$, Co$_3$O$_4$, NiO) catalysts were studied to investigate the effect of the dual loading of transition metals and CuO on the catalytic activity. As shown in Figure 7a, it can be observed that the CO conversion gradually increases with the increase of the reaction temperature until it reached 100%. As can be observed, the temperature of the 90% CO conversion over the 10CuO-3Co$_3$O$_4$/$\alpha$-MnO$_2$-400 catalyst was 75 °C. The activity of the 10CuO-3NiO/$\alpha$-MnO$_2$-400 catalyst basically shows a similar catalytic activity to that of the pristine 10CuO/$\alpha$-MnO$_2$-400 catalyst without modification (T$_{90}$ = 77 °C). However, the CO oxidation activity of the 10CuO-3Fe$_2$O$_3$/$\alpha$-MnO$_2$-400 catalyst (T$_{90}$ = 80 °C) was even worse than that of the pristine 10CuO/$\alpha$-MnO$_2$-400 catalyst. Therefore, only the catalytic activity of the 10CuO-3Co$_3$O$_4$/$\alpha$-MnO$_2$-400 was significantly improved compared with the pristine 10CuO/$\alpha$-MnO$_2$-400 catalyst. The reason for this might be that the loading of Cu and Co could generate more oxygen vacancies and active sites to the $\alpha$-MnO$_2$ nanowire. From the order of catalyst activity, it can be observed that the catalytic activity of the catalyst increases with the increase in of Mn$^{3+}$ content. Meanwhile, the loading of Co$_3$O$_4$ increased the specific surface area and pore volume of the catalyst, providing more active sites for the reaction. The ratio of O$_{ads}$/O$_{latt}$ was not exactly the same as the catalytic activity of the catalyst. The reasons for this phenomenon were stated in the discussion. The CO oxidation activity was significantly improved over the 10CuO-3Co$_3$O$_4$/$\alpha$-MnO$_2$-400 catalyst. This result was well consistent with the Mn 2p XPS analysis. The results of the normalized reaction rates of the four catalysts are shown in Figure 7b. It can be observed that the normalized reaction rates gradually increase with the increase of the reaction temperature. The order of reaction rates of the catalysts per surface area was 10CuO-3NiO/MnO$_2$ > 10CuO/MnO$_2$ > 10CuO-3Co$_3$O$_4$/MnO$_2$ > 10CuO-3Fe$_2$O$_3$/MnO$_2$. The normalized reaction rates excluded the effect of specific surface area on catalytic activity and expressed the intrinsic catalytic activity of the catalysts. The order of the O$_{ads}$/O$_{latt}$ ratios was consistent with the order of the normalized reaction rates.

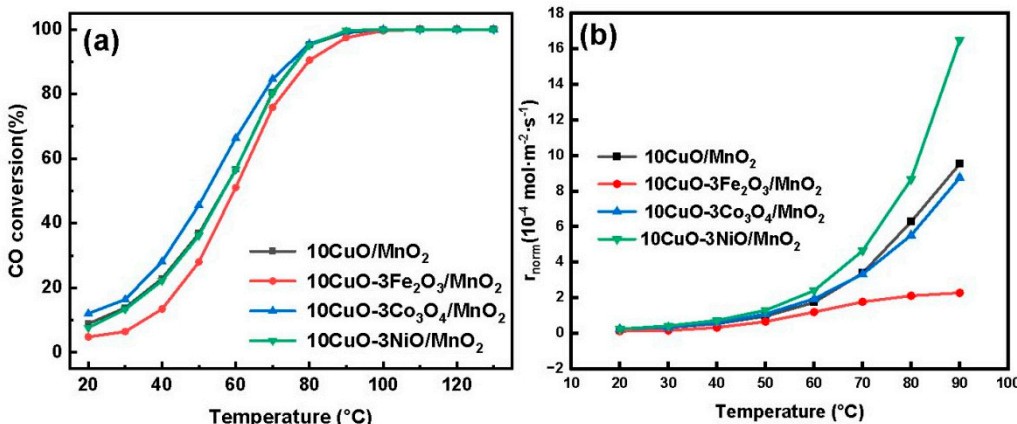

**Figure 7.** (**a**) CO conversions and (**b**) normalized reaction rates over the 10CuO-3MOx/α-MnO$_2$-400 (MOx = Fe$_2$O$_3$, Co$_3$O$_4$, NiO) catalysts under the reaction conditions: CO/O$_2$/N$_2$ = 1/20/79, GHSV = 12,000 mL·g$^{-1}$·h$^{-1}$, 1 atm.

### 2.6.2. The Effect of the Co$_3$O$_4$ Loading Amount on the Activities

The effect of Co$_3$O$_4$ loading on the catalytic activity of the CO oxidation was investigated, and the profiles of the CO conversion are shown in Figure 8a. It could be noticed that the Co$_3$O$_4$ loading amount in the range of 0 wt.%–10 wt.% had little effect on the CO oxidation activity of the 10CuO/α-MnO$_2$ catalyst. The order of CO catalytic activity of the catalysts is 10CuO-3Co$_3$O$_4$/α-MnO$_2$-400(T$_{90}$ = 75 °C) > 10CuO-10Co$_3$O$_4$/α-MnO$_2$-400 (T$_{90}$ = 77 °C) ≈ 10CuO/α-MnO$_2$-400 > 10CuO-1Co$_3$O$_4$/α-MnO$_2$-400 (T$_{90}$ = 79 °C) ≈ 10CuO-5Co$_3$O$_4$/α-MnO$_2$-400 > 10CuO-7Co$_3$O$_4$/α-MnO$_2$-400 (T$_{90}$ = 84 °C). The 10CuO-3Co$_3$O$_4$/α-MnO$_2$-400 catalyst performed the highest activity in the low temperature region. It is shown that a certain increase in the loading of Co$_3$O$_4$ was beneficial to the catalytic activity of the catalyst.

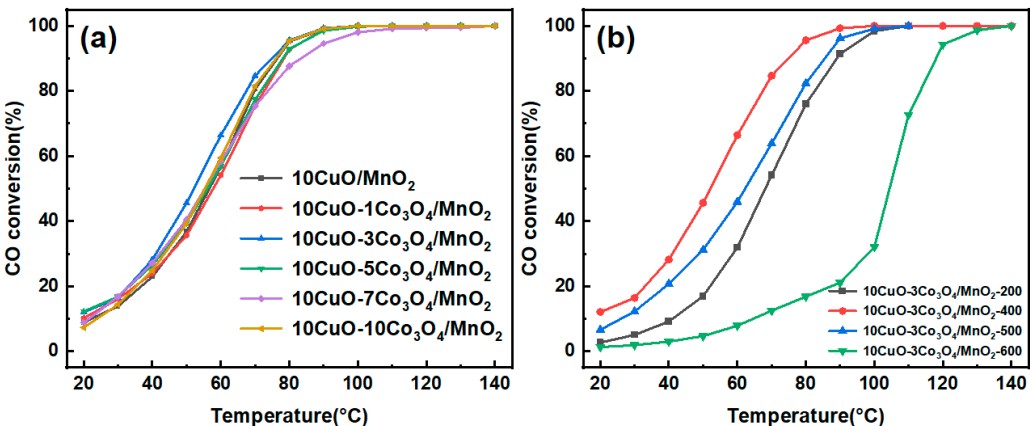

**Figure 8.** CO conversions over the (**a**) 10CuO-yCo$_3$O$_4$/α-MnO$_2$-400 (y = 0, 1, 3, 5, 7, 10), and (**b**) 10CuO-3Co$_3$O$_4$/α-MnO$_2$-T (T = 200, 400, 500, 600 °C) catalysts under the reaction conditions: CO/O$_2$/N$_2$ = 1/20/79, GHSV = 12,000 mL·g$^{-1}$·h$^{-1}$, 1 atm.

### 2.6.3. The Effect of the Calcination Temperature on the Activities

The catalytic CO oxidation of the 10CuO-3Co$_3$O$_4$/α-MnO$_2$-T (T = 200, 400, 500, 600 °C) catalyst was conducted to study the influence of the calcination temperature on catalytic activity, and the CO conversion profiles are shown in Figure 8b. The order of CO catalytic activity of catalysts is 10CuO-3Co$_3$O$_4$/α-MnO$_2$-400 (T$_{90}$ = 75 °C) > 10CuO-3Co$_3$O$_4$/α-MnO$_2$-500 (T$_{90}$ = 86 °C) > 10CuO-3Co$_3$O$_4$/α-MnO$_2$-200 (T$_{90}$ = 89 °C) > 10CuO-3Co$_3$O$_4$/α-MnO$_2$-600 (T$_{90}$ = 118 °C). It can be observed that catalytic activity has significantly decreased with the increase of the calcination temperature from 400 °C to 600 °C, especially over

the 10CuO-3Co$_3$O$_4$/α-MnO$_2$-600 catalyst. Specifically, the 10CuO-3Co$_3$O$_4$/α-MnO$_2$-600 catalyst had low activity of CO oxidation in the low temperature region. The possible reason is that the formation of a copper manganese spinel in the catalyst after calcination at high temperatures led to the significant decrease in the CO adsorption and oxidation capacities of the catalysts. This result is consistent with the XRD analysis. The 10CuO-3Co$_3$O$_4$/α-MnO$_2$-200 catalyst was also prepared for comparison. The results show that the catalytic activity of the 10CuO-3Co$_3$O$_4$/α-MnO$_2$-200 catalyst was significantly lower than that of the 10CuO-3Co$_3$O$_4$/α-MnO$_2$-400 catalyst. The possible reason is that the precursor of the Co$_3$O$_4$ could not be completely decomposed at 200 °C.

### 2.6.4. Stability Tests

The 12 h stability measurement was conducted over the 10CuO-3Co$_3$O$_4$/α-MnO$_2$-400 catalyst under the specific conditions (80 °C, CO/O$_2$/N$_2$ = 1/20/79, GHSV = 12,000 mL·g$^{-1}$·h$^{-1}$, 1 atm), and the result is shown in Figure 9a. It can be observed that the initial activity of the 10CuO-3Co$_3$O$_4$/α-MnO$_2$-400 catalyst can achieve 100% CO conversion in the first 2 h. Then, the CO conversion gradually decreased from 100% to 80% in the subsequent 2 h test. This indicates that the catalyst began to deactivate. However, the CO conversion could remain stable in the subsequent 8 h. This suggests that the 10CuO-3Co$_3$O$_4$/α-MnO$_2$-400 catalyst was provided with relatively good stability to some degree.

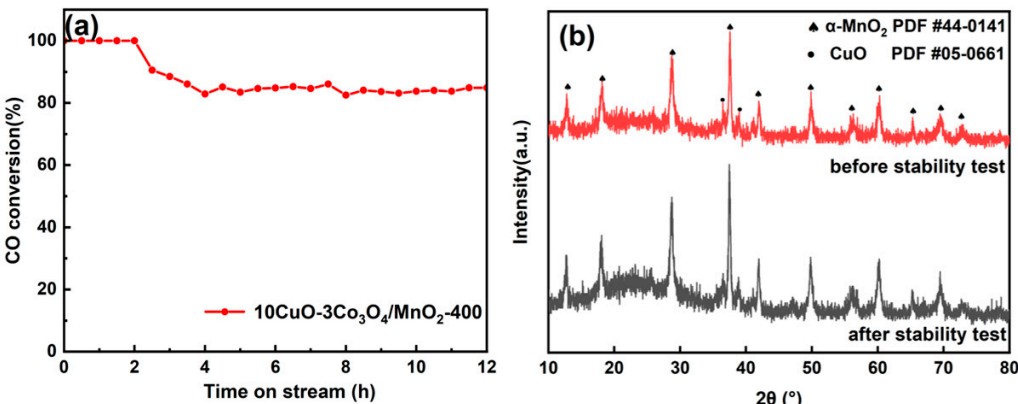

**Figure 9.** (**a**) The 12 h stability test of CO oxidation over the 10CuO-3Co$_3$O$_4$/α-MnO$_2$-400 catalyst under the conditions: 80 °C, CO/O$_2$/N$_2$ = 1/20/79, GHSV = 12,000 mL·g$^{-1}$·h$^{-1}$, 1 atm; (**b**) The XRD patterns of the fresh and the spent 10CuO-3Co$_3$O$_4$/α-MnO$_2$-400 catalysts before and after the stability test.

The XRD pattern of the 10CuO-3Co$_3$O$_4$/α-MnO$_2$-400 catalyst after the 12 h stability test is shown in Figure 9b. It can be observed that the diffraction peaks of the spent 10CuO-3Co$_3$O$_4$/α-MnO$_2$-400 catalyst after the 12 h stability test were a bit narrower and sharper than the fresh catalyst before the stability test. The possible reason is that the 10CuO-3Co$_3$O$_4$/α-MnO$_2$-400 catalyst underwent a bit of thermal agglomeration of the CuO active sites and the α-MnO$_2$ nanowire support during the catalytic process due to the hot spots of the catalyst bed, which could partly reduce the stability of the catalyst.

### 3. Discussion

The oxidation of CO on Cu-doped MnO$_2$ follows the Mars–van Krevelen (MvK) mechanism [35]. The reaction is divided into two parts: CO is first adsorbed on the catalyst surface and then reacts with surface-active oxygen on the catalyst surface to produce CO$_2$, which then generates oxygen vacancies on the catalyst surface. O$_2$ replenishes the depleted surface-active oxygen. After these two processes, the reaction completes a cycle [59]. There are redox electron pairs in the Cu-doped MnO$_2$ catalyst: -Cu$^{2+}$-O$^{2-}$-Mn$^{4+}$- →-Cu$^+$-□-Mn$^{3+}$- + 1/2O$_2$ (□ indicates the oxygen vacancy) [45]. The content of Mn$^{3+}$ on the MnO$_2$ catalyst is higher, presumably with more oxygen vacancies on the MnO$_2$ [60].

The order of the oxygen vacancy content of the catalysts is $10CuO-3Co_3O_4/\alpha-MnO_2 >$ $10CuO-3NiO/\alpha-MnO_2 > 10CuO/\alpha-MnO_2 > 10CuO-3Fe_2O_3/\alpha-MnO_2$. The oxygen vacancy content is consistent with the $Mn^{3+}$ content and catalyst activity. $O_2$ is commonly activated near the oxygen vacancy, producing surface active oxygen species ($O_{sur}$) [61]. It is well known that the higher the surface oxygen vacancy, the more easily $O_2$ is activated to reactive oxygen species [50]. However, the XPS spectra of O 1s showed that the order of the $O_{ads}/O_{latt}$ ratios was not consistent with the oxygen vacancy content and catalyst activity. To evaluate the intrinsic activity of the catalysts, the surface area normalized reaction rates are determined. The results of the surface area normalized reaction rates show that the loading of NiO has the greatest effect on the intrinsic activity of the catalyst. The $10CuO-3NiO/\alpha-MnO_2$ catalyst did not exhibit superior catalytic activity because the specific surface area of the catalyst after NiO loading was reduced, and the effect of specific surface area on the activity of the CO oxidation reaction could not be ignored. The $10CuO$-$3NiO/\alpha-MnO_2$ catalyst has the most surface adsorbed oxygen and reaction rates per unit surface area, but the small specific surface area results in the catalytic activity being similar to that of the pristine $10CuO/\alpha-MnO_2$-400 catalyst. The order of the intrinsic activity of the catalyst is consistent with the order of the $O_{ads}/O_{latt}$ ratio ($10CuO-3NiO/\alpha-MnO_2 >$ $10CuO/\alpha-MnO_2 > 10CuO-3Co_3O_4/\alpha-MnO_2 > 10CuO-3Fe_2O_3/\alpha-MnO_2$). This suggests that the surface-adsorbed oxygen is the reactive oxygen species involved in the oxidation of CO. $CO_2$ was produced by CO reacting with surface-adsorbed oxygen species [62,63]. After loading different transition metals, the catalysts form different types of oxygen vacancies, which have different electron densities and affect the production of reactive oxygen species [39]. The oxygen vacancies of $10CuO-3Co_3O_4/\alpha-MnO_2$ did not activate $O_2$ as well, and the $10CuO-3Co_3O_4/\alpha-MnO_2$ catalyst did not form more surface adsorbed oxygen. This might be the cause of the $10CuO-3Co_3O_4/\alpha-MnO_2$ catalyst providing the highest oxygen vacancy but poor surface-adsorbed oxygen. Although the normalized reaction rate of the $10CuO-3Co_3O_4/\alpha-MnO_2$ catalyst is not the highest, its high specific surface area allows for a greater number of oxygen vacancies. The large number of oxygen vacancies of the $10CuO-3Co_3O_4/\alpha-MnO_2$ catalyst counteracted the low activity of the oxygen vacancies and performed the high catalytic activity of CO oxidation. Therefore, the $10CuO-3Co_3O_4/\alpha-MnO_2$ catalyst exhibited the highest activity owing to its maximum specific surface area. The activity of the $10CuO-3Co_3O_4/\alpha-MnO_2$ catalyst was slightly higher than that of the $10CuO-3NiO/\alpha-MnO_2$ catalyst.

## 4. Materials and Method

### 4.1. Synthesis of $\alpha-MnO_2$ Nanowire Supports

The $\alpha-MnO_2$ nanowire support was facilely prepared via the typical hydrothermal method. Typically, 2 mmol $KMnO_4$ (Sinopharm Chemical Reagent Co., Ltd., Shanghai, China, AR, >99.5%) and 3 mmol $MnSO_4 \cdot H_2O$ (Shanghai Aladdin Biochemical Technology Co., Ltd., Shanghai, China, AR, 99%) were absolutely dissolved in 40 mL deionized water and vigorously stirred for 5 min, respectively. Then, these two solutions were mixed together by adding the $KMnO_4$ solution into the $MnSO_4$ solution. After stirring for 30 min, the obtained suspension was transferred to the 100 mL Teflon-lined stainless-steel autoclave. The autoclave was kept at 160 °C for 12 h. The obtained precipitate after the hydrothermal reaction was separated by the centrifuge and washed with ethanol (Sinopharm Chemical Reagent Co., Ltd., Shanghai, China, AR) and deionized water for three times. Then, the final powder was dried in the 100 °C oven overnight after the centrifugation. The obtained $\alpha-MnO_2$ was used as a support for subsequent experiments.

### 4.2. The Preparation of the Transition Metal Oxide (Fe₂O₃, Co₃O₄, NiO)-Doped CuO-Based Catalysts Supported on the α-MnO₂ Nanowire

The transition metal oxides promoted CuO-based $\alpha-MnO_2$ nanowire supported catalysts were prepared by the deposition precipitation method as reported in our previous work [64]. The weight percentages of the CuO and transition metal oxides were controlled

at x% and y% (x% = $m_{CuO}/(m_{CuO} + m_{MOx} + m_{support}) \times 100\%$; y% = $m_{MOx}/(m_{CuO} + m_{MOx} + m_{support}) \times 100\%$) by using the $Cu(NO_3)_2 \cdot 6H_2O$ (Shanghai Xinbao Fine Chemical Industry Factory, Shanghai, China, AR, >99.5%), $Fe(NO_3)_3 \cdot 9H_2O$ (Shanghai Macklin Bio-Chem Co., Ltd., Shanghai, China, AR, 98.5%), $Co(NO_3)_2 \cdot 6H_2O$ (Shanghai Macklin Bio-Chem Co., Ltd., Shanghai, China, AR, 99%), $Ni(NO_3)_2 \cdot 6H_2O$ (Sinopharm Chemical Reagent Co., Ltd., Shanghai, China, AR, 99%) as the precursors. For the specific procedure, the $\alpha$-$MnO_2$ nanowire was firstly dispersed in a $Cu(NO_3)_2 \cdot 6H_2O$ and $Fe(NO_3)_3 \cdot 9H_2O$/$Co(NO_3)_2 \cdot 6H_2O$/$Ni(NO_3)_2 \cdot 6H_2O$ aqueous solution Then, the $Na_2CO_3$ (0.01M, Shanghai Ling Feng Chemical Reagent Co., Ltd., Shanghai, China, AR, >99.8%) solution was added by droplet into the above solution to adjust the pH to 8–9 under vigorously stirring. The resultant solution was stirred for 30 min and then kept still for 1 h. The solid powder was separated by filtration and washed by the deionized water. The obtained catalyst was dried at 120 °C in an oven overnight and then calcinated at 400 °C for 5 h with a heating rate of 2 °C/min. The $\alpha$-$MnO_2$ nanowire-supported catalysts with 10 wt.% CuO and 3 wt.% MOx (MOx = $Fe_2O_3$, $Co_3O_4$, NiO) were denoted as the 10CuO-3MOx/$\alpha$-$MnO_2$ (MOx = $Fe_2O_3$, $Co_3O_4$, NiO). Meanwhile, the loading amount of the $Co_3O_4$ (wt.%) was subsequently changed in the same way. The obtained catalysts with loading amounts of 10 wt.% CuO and y wt.% $Co_3O_4$ were denoted as the 10CuO-y$Co_3O_4$/$\alpha$-$MnO_2$ (y = 0, 1, 3, 5, 7, 10). As for the influence of the calcination temperature, the catalysts with loading of the 10 wt.% CuO and 3 wt.% $Co_3O_4$ were calcinated at different temperatures and named 10CuO-3$Co_3O_4$/$\alpha$-$MnO_2$-T (T = 200, 400, 500, 600 °C).

### 4.3. Catalyst Characterizations

The X-ray diffraction (XRD) patterns of the catalysts were recorded on an X-ray power diffractometer (XRD-6100) from the Shimadzu Corporation by using the Cu K$\alpha$ rays (0.15046 nm), 40 KV tube voltage, and 40 mA tube current. The 2$\theta$ scanning range was from 10° to 80°, and the scanning speed was controlled at 3°/min. The scanning electron microscope (SEM) images were taken on an Apreo S Hivac instrument (Thermo Fisher Science, Waltham, MA, USA) with the accelerating voltage of 5 kV. The nitrogen physisorption was performed on an Autosorb-iQ-AG-MP instrument (Quantachrome, Boynton Beach, FL, USA) at liquid nitrogen temperature (−196 °C). The samples were degassed at 300 °C for 3 h to remove the surface-adsorbed water and impurities before the regular test. The specific surface areas of the catalysts were calculated by the Brunauer-Emmett-Teller (BET) method, and the pore size distribution and pore volume were calculated from the adsorption branch of the isotherm by the Barrett-Joyner-Halenda (BJH) method in the range of 0–1.0 P/P0. The X-ray photoelectron spectroscopy (XPS) measurements were performed on a Thermo Science K-Alpha + spectrometer (Thermo Fisher Science, Waltham, MA, USA). For the XPS measurement, the penetration depth of each catalyst was about 1–2 nm. The binding energy (BE) was calibrated by using C 1s = 284.8 eV as the standard. A fixed-bed reactor was used to conduct the hydrogen temperature-programmed reduction ($H_2$-TPR) experiment. The hydrogen consumption profile was recorded and analyzed with the online LC-D200 mass spectrometer (TILON GRP TECHNOLOGY LIMITED, Shanghai, China). The mixture of $H_2$ (1.2 mL/min) and Ar (23.7 mL/min) was introduced into the reactor. For each test, 100 mg of catalyst was loaded. After the hydrogen signal baseline line ($m/z$ = 2) was stabilized, the $H_2$-TPR test was performed with a heating rate of 20 °C/min from room temperature to 800 °C.

### 4.4. Catalyst Evaluation

The activity of the catalysts was evaluated in a fixed-bed reactor equipped with a quartz tube (i.d. = 8.00 mm). For each test, 100 mg catalyst was placed in the center of the quartz tube. The reactant gases, with a composition of 1% CO, 21% $O_2$, 79% $N_2$ (20 mL/min), were introduced into the reactor with the gas hourly space velocity (GHSV) of 12,000 mL·g$^{-1}$·h$^{-1}$. The catalytic activities of CO oxidation over different catalysts were tested in the specified temperature range. Each catalyst was tested three times at each

temperature. The outlet gases were analyzed online by using the GC-680 Perkin Elmer gas chromatograph equipped with the thermal conductivity detector (TCD). The 24 h stability tests of catalysts were carried out at 80 °C with the GHSV of 12,000 mL·g$^{-1}$·h$^{-1}$.

The catalytic performances of CO oxidation over these catalysts were stated in the form of the CO conversion. The calculated formula is listed below:

$$X_{CO} = (C_{CO,\ Inlet} - C_{CO,\ Outlet})/C_{CO,\ Inlet} \times 100\% \tag{1}$$

where $X_{CO}$ is the conversion rate of CO; and $C_{CO,\ Inlet}$ (ppm), and $C_{CO,\ Outlet}$ (ppm) are CO flowing into and out of the reactor, respectively.

In order to evaluate the intrinsic rate of CO oxidation on these catalysts, the calculated formula of the specific surface area normalization reaction rate is listed below [25]:

$$r_{norm}\left(mol \cdot m^{-2} \cdot s^{-1}\right) = \frac{C_{inlet} \cdot F}{m_{cat} \cdot S_{BET}} \cdot \ln\left(\frac{1}{1 - X_{CO}}\right) \tag{2}$$

where $r_{norm}$ (mol·m$^{-2}$·s$^{-1}$) is the normalized reaction rate, F (mol·s$^{-1}$) is the CO flow rate, $m_{cat}$ (g) is the mass of catalyst, and $S_{BET}$ (m$^2$·g$^{-1}$) is the BET surface area.

## 5. Conclusions

In this work, the novel $\alpha$-MnO$_2$ nanowire was prepared by the one-step hydrothermal method. A series of transition metal oxides (Fe$_2$O$_3$, Co$_3$O$_4$, NiO) promoted the CuO-based catalyst supported on the $\alpha$-MnO$_2$ nanowire and were synthesized by the co-precipitation method for the CO oxidation reaction. The effects of the transition metal oxide type, the loading amount, and the calcination temperature on the CO oxidation reaction had been systematically investigated. It was found that the 10CuO-3Co$_3$O$_4$/$\alpha$-MnO$_2$-400 catalyst showed the highest reactivity with T$_{90}$ = 75 °C. It was found that the 10CuO-3Co$_3$O$_4$/$\alpha$-MnO$_2$-400 catalyst possessed the largest specific surface area and exposed more active sites, which could further enhance the catalytic activity. The 10CuO-3NiO/$\alpha$-MnO$_2$-400 catalyst had the highest surface-adsorbed oxygen content and normalized reaction rate. This indicated that the surface-adsorbed oxygen was the surface-active oxygen involved in the oxidation of CO. It was found that the 10CuO-3Co$_3$O$_4$/$\alpha$-MnO$_2$-400 catalyst suffered from some deactivation during the 12 h stability test, which might be caused by the thermal sintering and agglomeration of the CuO active sites and $\alpha$-MnO$_2$ nanowire support. This should be the key consideration when designing the CuO-based CO oxidation catalyst in the future.

**Author Contributions:** Investigation, H.Z., H.S., Y.Z., Y.C., C.-e.W. and J.Q.; formal analysis, H.Z., H.S., Y.Z., Y.C., Y.X., C.-e.W. and J.X.; conceptualization, H.Z., Y.Z., H.S. and J.Q.; writing—original draft preparation, Y.Z. and L.X.; writing—review and editing, C.P., L.X. and M.C.; supervision, C.P., L.X. and M.C.; funding acquisition, L.X. and M.C.; project administration, L.X. and M.C. All authors have read and agreed to the published version of the manuscript.

**Funding:** This research was funded by the National Natural Science Foundation of China (Grant Nos. 22276098, 21976094, and 22176100), the National Key Research and Development Project (Grant No.2018YFC0213802), the Jiangsu Province "Carbon Peak and Carbon Neutrality Science and Technology Innovation Special Fund (The Third Batch)—Industry Foresight and Key Core Technology Research (Grant No. BE2022033-2), and the Postgraduate Research and Practice Innovation Program of Jiangsu Province (SJCX22_0367 and KYCX22_1216).

**Data Availability Statement:** The data presented in this study are available on request from the corresponding author. The data are not publicly available due to privacy.

**Conflicts of Interest:** The authors declare no conflict of interest.

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
