# Peer review of "Transition Metal (Fe2O3, Co3O4 and NiO)-Promoted CuO-Based α-MnO2 Nanowire Catalysts for Low-Temperature CO Oxidation"

_catalysts, doi:10.3390/catal13030588_

Round 1

Reviewer 1 Report

In the manuscript transition metals (Fe2O3, Co3O4 and NiO) promoted CuO-based α-MnO2 nanowire catalysts were studied for low-temperature CO oxidation. the CuO/α-MnO2 catalyst modified with 3 wt% Co3O4 and calcined at 400 °C performed the highest CO catalytic activity (T90 = 75 °C) among the investigated 24 catalysts. It was shown that the loading of the Co3O4 dopant increased the content of oxygen vacancies in the catalyst, the specific surface area and pore volume of the CuO/α-MnO2 nanowire catalyst, which would further enhance the catalytic activity. This work is relatively well-organized, a revision is needed before its publication. Here are some of my concerns.

1.    There are some errors or mistakes in the text. Please check carefully and correct them.

For example, there is an incorrect phrase: “This indicated that the oxidation activity of CO significantly depended on the phase structure and channel structure of the MnO2.” Activity of catalysts, not of CO!

2.    It is not clear how this can be seen from the figure 2: «As shown in Figure 2, the Cu and the doped Fe/Co/Ni elements were homogenously distributed over these investigated catalysts…..».  Perhaps the authors should give SEM-EDS data?

3.    It is necessary to explain why the different values (Fig.3a) ​ in the nitrogen volume adsorbed and desorbed by the samples starting from p/p<0,4. 

4.    It is not clear why the authors give the TPR-H2 data, if the samples were not previously reduced and the catalytic reaction temperature is 25-100 C. Please, give an explanation of why this method is in this investigation

5.    To discuss catalyst’s deactivation, XPS data after catalysis should be provided 

6.    Following on from the previous question, what caused the deactivation. What products can be formed by CO oxidation? What were the outlet gasesDid the authors check it?

7.    Please include a simple graphical abstract of catalytic active site of the catalyst. 

Author Response

In the manuscript transition metals (Fe2O3, Co3O4 and NiO) promoted CuO-based α-MnO2 nanowire catalysts were studied for low-temperature CO oxidation. the CuO/α-MnO2 catalyst modified with 3 wt% Co3O4 and calcined at 400 °C performed the highest CO catalytic activity (T90 = 75 °C) among the investigated 24 catalysts. It was shown that the loading of the Co3O4 dopant increased the content of oxygen vacancies in the catalyst, the specific surface area and pore volume of the CuO/α-MnO2 nanowire catalyst, which would further enhance the catalytic activity. This work is relatively well-organized, a revision is needed before its publication. Here are some of my concerns.

Answer: Thank you very much for providing such constructive comments and valuable suggestions to us. Your valuable comments are greatly important to improve the quality of the manuscript. The authors have made the best efforts to revise the whole manuscript according to your valuable and inspiring comments. Please kindly check the revised manuscript as a reference. The authors sincerely wish that the revised manuscript and the corresponding explanations could make you satisfied. Any further suggestions and comments are also greatly welcome and highly appreciated. Thank you.

  1. There are some errors or mistakes in the text. Please check carefully and correct them. For example, there is an incorrect phrase: “This indicated that the oxidation activity of CO significantly depended on the phase structure and channel structure of the MnO2.” Activity of catalysts, not of CO!

Answer: Thank you very much for pointing out this problem in the manuscript. Following your valuable suggestions, the authors have devoted our best efforts to revising the whole manuscript. Please kindly check the revised manuscript as the reference. Thank you.

  1. It is not clear how this can be seen from the figure 2: «As shown in Figure 2, the Cu and the doped Fe/Co/Ni elements were homogenously distributed over these investigated catalysts…..».  Perhaps the authors should give SEM-EDS data?

Answer: Thank you very much for your constructive comments. According to your comments, the authors have modified Figure 2. Please kindly find the revised manuscript as a reference. Thank you.

  1. It is necessary to explain why the different values (Fig.3a) in the nitrogen volume adsorbed and desorbed by the samples starting from p/p0 <0,4. 

Answer: Thank you for your valuable question. The authors believe that this result may have been caused by a problem with the air tightness of the device due to the operation of the measurement of the nitrogen adsorption-desorption isotherms. In order to correct this measurement error, the authors have re-characterized the nitrogen adsorption-desorption isotherms of 10CuO-3Co3O4/MnO2 catalyst and have revised the relevant graphs in the manuscript based on the second results. The authors have revised and refined the relevant formulation in the revised manuscript. Please kindly check the revised manuscript as the reference. Thank you.

  1. It is not clear why the authors give the TPR-H2 data, if the samples were not previously reduced and the catalytic reaction temperature is 25-100 C. Please, give an explanation of why this method is in this investigation.

Answer: Thank you very much for your constructive comments. H2-TPR is an effective tool to investigate the reducibility of metal oxide-based catalysts. In this work, the authors used the H2-TPR to study the metal-support interaction between the MnO2 support and the loading metal. The reduction peaks obtained at different temperatures indicated the presence of the different CuO species with various metal-support interactions. The authors carefully compared the reduction temperatures between the catalysts modified with the transition metal oxides and the pristine counterpart without loading the transition metal oxides. The authors sincerely wish that our explanation could make you clear. Thank you.

  1. To discuss catalyst’s deactivation, XPS data after catalysis should be provided.

Answer: Thank you very much for your inspiring and sensible suggestion. Your valuable suggestion is very reasonable. Unfortunately, the authors cannot successfully find the catalysts after the long-term stability reaction. The spent catalysts might be accidentally lost due to the renovation of the laboratory. Therefore, the authors are sincerely sorry that the authors cannot carry out the XPS analysis and provide the XPS data of the spent catalysts in this condition. Thank you so much for your kind understanding.

  1. Following on from the previous question, what caused the deactivation. What products can be formed by CO oxidation? What were the outlet gases? Did the authors check it?

Answer: Thank you very much for your attention to our manuscript. In the manuscript, the authors used the conversion rate of CO to assess the activity of the catalyst. The outlet gases were analyzed online by using the GC-680 Perkin Elmer gas chromatograph equipped with the thermal conductivity detector (TCD). It could be observed that only two characteristic peaks of CO and CO2 appeared on the TCD spectrum of the outlet gases. Therefore, it could be concluded that CO is oxidized to CO2 on the catalyst and the final product of the CO oxidation is CO2. The authors sincerely wish that our explanation can make you satisfied and clear. Thank you.

  1. Please include a simple graphical abstract of catalytic active site of the catalyst. 

Answer: Thank you very much for your sensible suggestion. According to your valuable suggestions, a simple graphical abstract was shown in the revised manuscript submission. Thank you.

Reviewer 2 Report

Comments are attached. 

Author Response

Reviewer #2

The current paper deals with the addition of transition metals (Fe, Co, Ni) to CuO-based MnO2 nanowire catalysts for CO oxidation. The study uses various characterization techniques and catalytic activity measurements to identify the best catalyst for the reaction. While the authors identify that the addition of Co3O4 is most effective in lowering the temperature for the conversion of CO to CO2, the underlying the reasons for the improvement in catalytic activity (oxygen vacancy formation energy) are not supported by the evidence presented from the characterization methods or the references cited. Moreover, it is not clear as to how the oxygen vacancy formation energy is relevant in the discussion of the overall reaction mechanism. I would recommend that the authors reconsider the writing in the discussion based on the comments below.

Answer: Thank you very much for your selfless dedication and providing such inspiring, constructive, and sensible comments for us. Your valuable comments are of great significance for us to improve the quality of the manuscript. Following your valuable suggestions, we have devoted our best efforts to revising the whole manuscript. Please kindly check the revised manuscript as the reference. We sincerely wish that our revised manuscript can make you satisfied. Any further suggestions as well as comments are greatly welcome and appreciated.

  1. Page 2, Line 63: It was believed that the catalytic activity of MnO2 was greatly related to the oxygen vacancy activity [29].The authors need to discuss the relation between catalytic activity and oxygen vacancy formation more explicitly.

Answer: Thank you very much for your attention to our manuscript. In reference [29], Tian et al. oxidized CO with different crystalline phases of MnO2 and determined the oxygen vacancy content of different crystalline phases of MnO2 by in situ electron paramagnetic resonance (EPR) spectra, showing that the order of oxygen vacancy content was proportional to the order of catalytic activity (The order of oxygen vacancy content was β-MnO2>ε-MnO2>α-MnO2 and the order of catalytic activity was β-MnO2>ε-MnO2>α-MnO2). Oxygen vacancies are seen as important active sites for the adsorption and dissociation of oxygen molecules, so the catalytic activity of the catalyst is dependent on the concentration of oxygen vacancies. Thank you very much for your kindly suggestions and corrections, we have made changes to the relevant parts of the manuscript. For more information on the formation of oxygen vacancies please refer to the added discussion section. Please kindly find the revised manuscript as a reference. Thank you.

  1. Page 3, Line 105: “2.1. Synthesis of α-MnO2 Nanowire Supports.” Please include the

source and purity of the chemicals used for the study.

Answer: Thank you very much for your constructive suggestion. The authors have added the source and purity of the chemicals in sections 2.1 and 2.2. Please kindly find the revised manuscript as a reference. Thank you.

  1. Page 5, Line 203: “Meanwhile, the intensity of diffraction peaks of CuO became sharp with the increase of the calcination temperature.” This is not clear from Figure 1c. A sharp peak emerging at ~38 º with increasing temperature is labelled as copper manganese oxide but the CuO peak at ~40º does not appear to grow. Please double check the XRD peak assignments.

Answer: Thank you for pointing out this unclear statement in manuscript. The authors indicated the position of copper-manganese spinel in Figure 1(c). It can be observed that the diffraction peak of Cu-Mn spinel became stronger with increasing calcination temperature. The powder XRD pattern can only clearly observe one diffraction peak (38.8°) of CuO after loading the Co3O4, the diffraction peak of CuO on the catalyst was enhanced after calcination from 200°C to 400 °C. We are very sorry for our negligence of this unclear statement, which make you feel confused. Following your advice, the authors have revised and refined the relevant formulation in the revised manuscript. Please kindly check the revised manuscript as the reference. Thank you.

  1. Page 5, Line 211”: “(T = 200, 400, 500, 600)”: Please include ºC in the figure caption

here and everywhere else where relevant.

Answer: Thank you very much for your constructive comments. According to your comments, the authors have corrected the errors and examined in detail the full text. Please kindly find the revised manuscript as a reference. Thank you.

  1. Page 5, Line 219: “The reason for 219 this was caused by the high loading contents of the CuO and transition metal oxides” It is not clear as to why CuO and transition metal loading would lead to a decrease in the length to diameter ratio of the MnO2 nanowires. The authors need to add an explanation for this.

Answer: Thank you very much for your valuable and inspiring comments. Indeed, this phenomenon has also been identified and reported in the previous work (Nanomaterials 2022; 12:2083.). However, there was no explanation provided in this work. The authors plan to discover the causes of this phenomenon in the subsequent work in the future with the help of other characterizing techniques. Thank you so much for your kind understanding.

  1. Page 5, Line 225: “As shown in Figure 2, the Cu and the doped Fe/Co/Ni elements were homogenously distributed over these investigated catalysts”: The image for Cu EDS mapping in Figure 2 is not clearly seen. Please correct this.

Answer: Thank you very much for pointing out this drawback in the manuscript. Following your valuable suggestions, the authors have devoted our best efforts to revising the whole manuscript and carefully modify the size of the image in the revised manuscript to make the EDS mapping seen clearly. Please kindly check the revised manuscript as the reference. Thank you.

  1. Page 8, Line 280: “The content of Mn3+ and oxygen vacancies was proportional, or indirectly proportional to the oxidation capacity of the catalyst [29]. The Mn3+ may cause the Jahn-Teller effect, which could prolong the Mn-O bond in [MnO6] [47-48], thereby prolonging the distance between the oxygen pairs and causing the stretching of the Mn-O bond length. As a result, the Mn-O bond was easier to break and the mobility of oxygen became higher. Therefore, the released surface oxygen atoms were more likely to participate in the reaction, and thus promote the catalytic performance.” Here, the authors claim that the presence of Mn3+ can increase oxygen mobility, thereby increase catalytic activity. However, the authors also refer to [29], where the CO oxidation activity was inversely proportional to the oxygen vacancy formation activity. There seems to be a contradiction here that the authors need to address.

Answer: Thank you very much for your valuable and inspiring comments. Your concern is very reasonable. Actually, lots of researchers have studied the relationship between low-valent manganese ions and oxygen vacancies. The common method of determining the oxygen vacancy concentrations directly is the EPR spectra. The manganese oxide showed the symmetrical EPR peaks at g = 2.003 in EPR spectra, which were ascribed to the unpaired electrons in the oxygen vacancies of metal oxides. The concentration of the oxygen vacancies could be reflected by the signal intensity (Appl. Catal. B Environ. 2020; 264:118464; Appl. Catal. B Environ. 2022; 312:121387; ACS Catal. 2020; 10(19):11082-11098.). The reference [29] showed that the oxygen vacancy content (determined by EPR  spectroscopy) was proportional to the catalytic activity (The order of oxygen vacancy content was β-MnO2>ε-MnO2>α-MnO2 and the order of catalytic activity was β-MnO2>ε-MnO2>α-MnO2). Some researchers' studies have shown that the amount of low-valent manganese ions (Mn3+) in the manganese oxide was proportional to the concentration of oxygen vacancies and also proportional to the oxidation capacity of the catalysts of the MnO2 (Chemosphere 2022; 291: 135890; Appl. Catal. B Environ. 2020, 260, 118150; Appl. Surf. Sci. 2023, 618, 156643.). Therefore, the two statements in the manuscript are well consistent. The authors sincerely wish that our explanation could make you clear. Besides, the authors also make clearer descriptions in the revised manuscript. Please kindly find the revised manuscript as the reference. Thank you.

  1. Page 8, Line 292: “It could be observed that the binding energy of surface lattice oxygen (Olatt) shifted to higher binding energy with the addition of transition metals.” Please add the details of the peak position of the surface lattice oxygen to Table 2.

Answer: Thank you very much for your valuable comments. Following your constructive advice, the authors have showed the specific position of surface lattice oxygen at the O 1s position in Table 3. The authors further have labeled the displacement of the binding energy in Figure 4 (b). Please kindly find the revised manuscript as the reference. Thank you.

  1. Page 9, Lines 309-327: It is not clear as to why the authors discuss the XPS spectra of the added transition metals (Fe 2p, Ni 2p, Co 2p). The authors need to add the relevance of the observations made in this section in the context of the CO oxidation reaction mechanism and reaction sites.

Answer: Thank you very much for your constructive comments. As well known, the transition metals have a variety of valence states. In this work, the authors used the XPS spectroscopy to determine the valences of the transition metal oxides loaded on the catalyst. Meanwhile, the authors could not observe the diffraction peaks of the transition metals oxides based on the XRD analysis owing to the low loading amount. Therefore, the XPS profiles could also show the transition metal oxides (Fe2O3, NiO, Co3O4) had been successfully loaded on CuO-MnO2. The authors sincerely wish that our explanation could make your concern clear. Besides, the authors also make the corresponding explanation in the revised manuscript. Please kindly find the revised manuscript as the reference. Thank you.

  1. Table 3 is missing from the text. Please include this.

Answer: Thank you very much for your constructive comments. According to your comments, the authors have corrected the errors of number error of Table 3. Please kindly find the revised manuscript as the reference. Thank you.

  1. Page 10, Lines 369: “As can be observed, the temperature of the 90% CO conversion over the 10CuO-3Co3O4/α-MnO2-400 catalyst was 75 °C. The activity of 10CuO- 3NiO/α-MnO2-400 catalyst showed basically similar catalytic activity to that of the pristine 10CuO/α-MnO2-400 catalyst without modification (T90 = 77°C). However, the CO oxidation activity of the 10CuO-3Fe2O3/α-MnO2-400 catalyst (T90 = 80°C) was even worse than that of the pristine 10CuO/α-MnO2-400 catalyst. Therefore, only the catalytic activity of the 10CuO-3Co3O4/α-MnO2-400 was significantly improved compare with the pristine 10CuO/α-MnO2-400 catalyst.” The authors claim improvement for the Co3O4 addition due to a decrease of 2 °C in T90. The authors need to provide further evidence to show that the temperature difference is significant enough to support the claim and not that this different is within measurement error.

Answer: Thank you very much for your attention to our manuscript. Your concern is quite reasonable. As for the catalytic activity, the authors actually measured the conversion of CO for three times at each temperature and calculated the average of the results to minimize the effect of measurement error. The catalysts showed consistent activity at each temperature. For example, the order of CO catalytic activity of catalysts was 10CuO-3Co3O4/α-MnO2-400 > 10CuO-3NiO/α-MnO2-400 ≈ 10CuO-α-MnO2-400 > 10CuO-3Fe2O3/α-MnO2-400 at each temperature. The authors sincerely wish that our explanation could make your concern clear. Besides, the authors also make the corresponding explanation and statement in the revised manuscript. Please kindly find the revised manuscript as the reference. Thank you.

  1. Page 10, Lines 377: “From the order 377 of catalyst activity, it can be seen that the catalytic activity of the catalyst increases with the content of Mn3+ increases. Meanwhile, the loading of Co3O4 increased the specific surface area and pore volume of the catalyst, providing more active sites for the reaction”. The two reasons for improvement suggested here need to be clearly differentiated. The role of Mn3+ needs to be evaluated after calculating the surface area normalized the reaction rates from BET. Only then can both the effects be discussed.

Answer: Thank you very much for your inspiring and sensible suggestion. According to your comments, the authors calculated the specific surface area normalization reaction rates of four catalysts (10CuO-α-MnO2-400, 10CuO-3Fe2O3/α-MnO2-400, 10CuO-3Co3O4/α-MnO2-400, 10CuO-3NiO/α-MnO2-400) based on the method described in the precursory literature (Appl. Catal. B Environ. 2020, 260, 118150.). The normalized reaction rate (rnorm, mol·m-2·s-1) is derived from the following formula:

Where Cinlet (ppm) is the concentration of CO in inlet gas, F (mol·s-1) is the CO flow rate, mcat (g) is the mass of catalyst, SBET (m2·g-1) is the BET surface area, the XCO (%) is the CO conversion. The result of specific surface area normalization reaction rate was shown in Figure 3. (c). The authors used specific surface area normalization to evaluate the intrinsic activity of the catalyst. It can be observed that the results of reaction rates were inconsistent with the activity order of the catalyst, indicating that the effect of specific surface area on catalytic activity cannot be eliminated. Because the 10CuO-3Co3O4/α-MnO2-400 catalyst had the largest specific surface area, the catalyst exhibited the highest activity. The results of normalized reaction rate further illustrated the positive effect of surface adsorbed oxygen on catalytic activity. The authors sincerely wish that our explanation could make you clear. Besides, the authors also make corresponding explanation in the revised manuscript. Please kindly find the revised manuscript as the reference. Thank you.

Figure 3. (c) Normalized reaction rates of the 10CuO/α-MnO2-400 and 10CuO-3MOx/α-MnO2-400 (MOx = Fe2O3, Co3O4, NiO) catalysts.

  1. Page 10, Lines 377: “The ratio of Oads/Olatt was not exactly the same as the catalytic activity of the catalyst, indicated that the amount of adsorbed oxygen was not the decisive factor affecting the catalytic activity.” This result is rather surprising, and the authors need to discuss this further especially that in Line 284 the authors mention, “As a result, the Mn-O bond was easier to break and the mobility of oxygen became higher. Therefore, the released surface oxygen atoms were more likely to participate in the reaction, and thus promote the catalytic performance.”, implying that the surface oxygen should be correlated to Mn3+ and be beneficial for the reaction.

Answer: Thank you very much for your inspiring and sensible suggestion. In fact, the authors had noted this phenomenon before. The authors had repeatedly calculated the ratio of surface adsorbed oxygen to surface lattice oxygen using the XPSPEAK 41 and Origin 8.5 softwares. The ratios of Oads/Olatt were derived on the basis of respecting actual data. In order to clarify the reasons for this phenomenon, the authors have carefully discussed it in the Discussion section. The authors make corresponding explanation in the revised manuscript. Please kindly find the revised manuscript as the reference. Thank you.

  1. Page 11, Lines 392: “The 10CuO-3Co3O4/α-MnO2-400 catalyst performed the highest activity in the low temperature region. The reason for this might be due to the increased Co3O4 load amount, resulting in the reduced CuO dispersion on the α-MnO2 support.” The authors need to add why reduced CuO dispersion would lead to improved activity at low temperature.

Answer: Thank you very much for your constructive comments and pointing out the problem in manuscript. After the authors' reconfirmation and discussion, we found that the current experimental results were insufficient to support this conclusion. The reason for this might be due to the increased Co3O4 load amount, resulting in the reduced CuO dispersion on the α-MnO2 support. Therefore, the author will carry out the subsequent investigation why the different loadings of Co3O4 leading to different catalytic activities. Meanwhile, the authors have modified the relevant parts of the manuscript. The authors are very sorry for our negligence of this faulty expression. Please kindly find the revised manuscript as a reference. Thank you.

  1. Page 12, Line 445: “It was found that the 10CuO-3Co3O4/α-MnO2-400 catalysts possessed the largest specific surface area and most abundant oxygen vacancies, which could facilitate the rapid conversion of oxygen species and further enhance the catalytic activity.” The authors need to clearly mention about they arrive at the oxygen vacancy conclusion as they have not measured oxygen vacancy concentration.

Answer: Thank you very much for your inspiring and sensible suggestion. As well known, the EPR is an effective tool to precisely measure the oxygen vacancy concentration. But the authors cannot perform the EPR analysis due to the limitation of the present experimental conditions. Therefore, the authors are sincerely sorry that we cannot carry out the EPR analysis considering the conditional restriction. Thank you so much for your kind understanding. The results of some researchers suggested that the presence of Mn3+ favored the production of surface oxygen vacancies (Appl. Surf. Sci. 2023; 618:156643.). It could be described by the following formula based on the principle of electrical neutrality: Mn4++O2-→Mn3+/Mn2++1/2O2(g)+Ov (Ov indicates the oxygen vacancy) (Appl. Catal. B Environ. 2020; 260:118150.). In copper-manganese oxides, there is a charge transfer between the copper and manganese cations: -Cu2+-O2--Mn4+-→-Cu+-£-Mn3+-+1/2O2 (£indicates the oxygen vacancy) (Chem. Eng. J. 2019, 357, 258-268.) The higher concentration of Mn3+ implies the higher concentration of oxygen vacancy, which has an important vital function for the catalytic reaction (J. Hazard. Mater. 2020; 391:122181; J. Environ. Sci. 2021; 104:102-112.). The content of Mn3+ is indirectly proportional to the oxygen vacancy (Appl. Surf. Sci. 2022; 606:154846.). The authors used an indirect approach to measure the production of oxygen vacancies. We sincerely wish that our explanation can make you satisfied and clear. Thank you.

  1. The authors need to add a discussion section to explain why Co3O4 addition is most beneficial for CO oxidation by including the surface area normalized reaction rates, the relevance of Mn3+, surface oxygen species, oxygen vacancies in the context of the overall reaction mechanism/steps and compare it to the Ref [29] mentioned in the text.

Answer: Thank you very much for your inspiring and sensible suggestion. The oxidation of CO on Cu-doped MnO2 follows the Mars–van Krevelen (MvK) mechanism (J. Catal. 2016; 341:82-90.). The reaction is divided into two parts: CO is first adsorbed on the catalyst surface and then reacts with surface active oxygen on the catalyst surface to produce CO2, which then generates oxygen vacancies on the catalyst surface. O2 replenishes the depleted surface active oxygen, after these two processes, the reaction completes a cycle (J. Phys. Chem. C 2019; 124(1):701-708). There are redox electron pairs in the Cu-doped MnO2 catalyst: -Cu2+-O2--Mn4+-→-Cu+-£-Mn3+-+1/2O2 (£ indicates the oxygen vacancy) (Chem. Eng. J. 2019, 357, 258-268.). The content of Mn3+ on the CuO doped MnO2 catalyst is the highest, presumably with the most oxygen vacancies on the MnO2 (J. Environ. Sci. 2021; 104, 102–112.). The order of oxygen vacancy content of the catalysts was 10CuO-3Co3O4/α-MnO2>10CuO-3NiO/α-MnO2>10CuO/α-MnO2>10CuO-3Fe2O3/α-MnO2. The oxygen vacancy content was consistent with Mn3+ content and catalyst activity. O2 is activated near the oxygen vacancy, producing surface active oxygen species (Osur).(Chem. Eng. J. 2023; 451:138868). It is well known that the higher the surface oxygen vacancy, the more easily O2 is activated to reactive oxygen species (Appl. Catal. B Environ. 2020; 264:118464). But the XPS spectra of O 1s showed that the order of Oads/Olatt ratios was not consistent with oxygen vacancy content and catalyst activity. To evaluate the intrinsic activity of the catalysts, the surface area normalized reaction rates were determined. The results of the surface area normalized the reaction rates showed that the loading of NiO had the greatest effect on the intrinsic activity of the catalyst. The 10CuO-3NiO/α-MnO2 catalyst did not exhibit superior catalytic activity because the specific surface area of the catalyst after NiO loading was reduced and the effect of specific surface area on the activity of the CO oxidation reaction could not be ignored. The 10CuO-3NiO/α-MnO2 catalyst had the most surface adsorbed oxygen and reaction rates per unit surface area, but a small specific surface area result in its catalyst activity being similar to that of the pristine 10CuO/α-MnO2-400 catalyst. The order of the intrinsic activity of the catalyst was consistent with the order of the Oads/Olatt ratio (10CuO-3NiO/α-MnO2>10CuO/α-MnO2>10CuO-3Co3O4/α-MnO2> 10CuO-3Fe2O3/α-MnO2). This suggesting that the surface adsorbed oxygen was the reactive oxygen species involved in the oxidation of CO. CO2 was produced by CO reacting with surface adsorbed oxygen species (J. Ind. Eng. Chem, 2015; 25: 250-257.)(Appl. Surf. Sci. 2022; 606:154846.). After loading different transition metals, the catalysts formed different types of oxygen vacancies, which had different electron densities and affected the production of reactive oxygen species (Appl. Catal. B Environ. 2021; 285:119873.). The oxygen vacancies of 10CuO-3Co3O4/α-MnO2 did not activate O2 as well, the 10CuO-3Co3O4/α-MnO2 catalyst did not form more surface adsorbed oxygen. The normalized reaction rate of the 10CuO-3Co3O4/α-MnO2 catalyst was not the highest, but its high specific surface area allowed for a greater number of oxygen vacancies. The large number of oxygen vacancies of 10CuO-3Co3O4/α-MnO2 catalyst counteracted the low activity of the oxygen vacancies and led to a high activity of the catalyst. The 10CuO-3Co3O4/α-MnO2 catalyst exhibited the highest activity because of its maximum specific surface area. The activity of the 10CuO-3Co3O4/α-MnO2 catalyst was slightly higher than that of the 10CuO-3NiO/α-MnO2 catalyst. The authors sincerely wish that our explanation could make you clear. Besides, the authors also make a corresponding explanation in the revised manuscript. Please kindly find the revised manuscript as a reference. Thank you.

Round 2

Reviewer 2 Report

Thank you for making the changes.

Author Response

Response to the reviewers’ comments (Manuscript ID: catalysts-2229287)

Thank you for addressing the comments and making changes as necessary. The manuscript reads well and is better organized now. A few comments are detailed below based on the changes made:

Thank you very much for providing such constructive comments and valuable suggestions to us. Your valuable comments are greatly important to improve the quality of manuscript. The authors have made the best efforts to revise the whole manuscript according to your valuable and inspiring comments. Please kindly check the revised manuscript as the reference. The authors sincerely wish that the revised manuscript and the corresponding explanations could make you satisfied. Any further suggestions and comments are also greatly welcome and highly appreciated. Thank you.

Comment 1: Thank you for the clarification about the measurement being made thrice. I would recommend either showing the error bars in the light-off curves or report the average temperature difference for T90 with the error bar.

Answer: Thank you very much for providing such constructive comments and valuable suggestions to the authors. According to your comments, the authors have inserted the error bars in the below Figure. It could be observed that error for every test was very small, even negligible. In order to make the Figure 7(a) consistent with the Figure 8 in the form, the authors did not add the error bars in the revised manuscript. Please kindly find the below Figure as the reference. Thank you very much for your kind understanding. Any further suggestions and comments are greatly welcome and highly appreciated. Thank you.

Figure. CO conversions over the 10CuO-3MOx/α-MnO2-400 (MOx = Fe2O3, Co3O4, NiO) catalysts under given reaction conditions: CO/O2/N2 = 1/20/79, GHSV = 12000 mLg-1h-1, 1 atm.

Comment 2: Thank you for the calculation of the surface area normalized reaction rates. However, it is typical in plug-flow reactor measurements that reaction kinetics are calculated under low conversion conditions (‘differential regime’, typically < 10 % conversion). Please update Fig. 7b taking into consideration only low conversion regimes.

Answer: Thank you very much for providing such constructive comments and valuable suggestions to us. Your comments are quite reasonable. As regards the data in this work, the CO conversions of 10CuO-3MOx/α-MnO2-400 (MOx = Fe2O3, Co3O4, NiO) catalysts were close to or more than 10 % even at the lowest temperature (20 °C). In this work, the activity of the catalysts was evaluated in the fixed bed reactor equipped with the quartz tube (i.d. = 8.00 mm). For each test, 100 mg catalyst was placed in the center of the quartz tube. The thickness of catalyst in the quartz tube was about or less than the 5 mm. The influence of mass diffusion and heat transfer could be ignored during our test. Therefore, the authors believed that the normalized reaction rate could well express the intrinsic activity of the catalysts. The authors sincerely wish that our explanation can make you satisfied and clear. Thank you.
